# Shank promotes action potential repolarization by recruiting BK channels to calcium microdomains

Luna Gao[1,2], Jian Zhao[1,2], Evan Ardiel[1,2], Qi Hall[1], Stephen Nurrish[1,2], Joshua M Kaplan[1,2,3]*

[1]Department of Molecular Biology, Massachusetts General Hospital, Boston, United States; [2]Department of Neurobiology, Harvard Medical School, Boston, United States; [3]Program in Neuroscience, Harvard Medical School, Boston, United States

**Abstract** Mutations altering the scaffolding protein Shank are linked to several psychiatric disorders, and to synaptic and behavioral defects in mice. Among its many binding partners, Shank directly binds CaV1 voltage activated calcium channels. Here, we show that the *Caenorhabditis elegans* SHN-1/Shank promotes CaV1 coupling to calcium activated potassium channels. Mutations inactivating SHN-1, and those preventing SHN-1 binding to EGL-19/CaV1 all increase action potential durations in body muscles. Action potential repolarization is mediated by two classes of potassium channels: SHK-1/KCNA and SLO-1 and SLO-2 BK channels. BK channels are calcium-dependent, and their activation requires tight coupling to EGL-19/CaV1 channels. SHN-1's effects on AP duration are mediated by changes in BK channels. In *shn-1* mutants, SLO-2 currents and channel clustering are significantly decreased in both body muscles and neurons. Finally, increased and decreased *shn-1* gene copy number produce similar changes in AP width and SLO-2 current. Collectively, these results suggest that an important function of Shank is to promote microdomain coupling of BK with CaV1.

## Editor's evaluation

Mutations altering the scaffolding protein Shank are linked to several psychiatric disorders. Here the authors take advantage of *C. elegans* genetics and muscle physiology to demonstrate that Shank binds CaV1 voltage activated calcium channels and promotes CaV1 coupling to calcium activated potassium channels.

*For correspondence:
kaplan@molbio.mgh.harvard.edu

Competing interest: The authors declare that no competing interests exist.

## Introduction

Shank is a synaptic scaffolding protein (containing SH3, PDZ, proline-rich and SAM domains) (*Grabrucker et al., 2011*). Mammals have three Shank genes, each encoding multiple isoforms (*Jiang and Ehlers, 2013*). Several mouse Shank knockouts have been described but these mutants exhibit inconsistent (often contradictory) synaptic and behavioral defects (*Jiang and Ehlers, 2013*), most likely resulting from differences in the Shank isoforms impacted by each mutation. The biochemical mechanism by which Shank mutations alter synaptic function and behavior has not been determined.

In humans, Shank mutations and CNVs are linked to Autism Spectrum Disorders (ASD), schizophrenia, and mania (*Durand et al., 2007*; *Peça et al., 2011*). Haploinsufficiency for 22q13 (which spans the Shank3 locus) occurs in Phelan-McDermid syndrome (PMS), a syndromic form of ASD (*Phelan and McDermid, 2012*). PMS patients exhibit autistic behaviors accompanied by hypotonia, delayed speech, and intellectual disability (ID) (*Bonaglia et al., 2011*). Heterozygous inactivating Shank3

mutations are found in sporadic ASD and schizophrenia (*Durand et al., 2007*; *Peça et al., 2011*). A parallel set of genetic studies suggest that increased Shank3 function also contributes to psychiatric diseases. 22q13 duplications spanning Shank3 are found in ASD, schizophrenia, ADHD, and bipolar disorder (*Durand et al., 2007*; *Failla et al., 2007*; *Han et al., 2013*). A transgenic mouse that selectively over-expresses Shank3 exhibits hyperactive behavior and susceptibility to seizures (*Han et al., 2013*). Taken together, these studies suggest that too little or too much Shank3 can contribute to the pathophysiology underlying these psychiatric disorders.

Given its link to psychiatric disorders, there is great interest in determining how Shank regulates circuit development and function. Shank is highly enriched in the post-synaptic densities of excitatory synapses; consequently, most studies have focused on the idea that Shank proteins regulate some aspect of synapse formation or function. Through its various domains, Shank proteins bind many proteins (*Lee et al., 2011*; *Sakai et al., 2011*), thereby potentially altering diverse cellular functions. Shank proteins have been implicated in activity induced gene transcription (*Perfitt et al., 2020*; *Pym et al., 2017*), synaptic transmission (*Zhou et al., 2016*), synapse maturation (*Harris et al., 2016*), synaptic homeostasis (*Tatavarty et al., 2020*), cytoskeletal remodeling (*Lilja et al., 2017*), and sleep (*Ingiosi et al., 2019*). Each of these defects could contribute to the neurodevelopmental and cognitive deficits observed in ASD and schizophrenia.

Several recent studies suggest that an important function of Shank is to regulate the subcellular localization of ion channels. Shank mutations decrease the synaptic localization of NMDA and AMPA type glutamate receptors (*Peça et al., 2011*; *Won et al., 2012*). Other studies show that Shank proteins promote delivery of several ion channels to the plasma membrane, including HCN channels (*Yi et al., 2016*; *Zhu et al., 2018*), TRPV channels (*Han et al., 2016*), and CaV1 voltage activated calcium channels (*Pym et al., 2017*; *Wang et al., 2017*). Of these potential binding partners, we focus on CaV1 because human CACNA1C (which encodes a CaV1 α-subunit) is mutated in Timothy Syndrome (TS), a rare monogenic form of ASD (*Splawski et al., 2005*; *Splawski et al., 2004*), and polymorphisms linked to CACNA1C are associated with multiple psychiatric disorders (*Psychiatric Genomics, 2013*). For this reason, we asked how Shank regulates the coupling of CaV1 channels to their downstream effectors.

*C. elegans* has a single Shank gene, *shn-1*. The SHN-1 protein lacks an SH3 domain but has all other domains found in mammalian Shank proteins. Mammalian Shank proteins directly bind CaV1 channels through both the SH3 and PDZ domains (*Zhang et al., 2005*). We previously showed that the SHN-1 PDZ domain directly binds to a carboxy-terminal ligand in EGL-19/CaV1 (*Pym et al., 2017*). CaV1 channels are tightly coupled to multiple downstream calcium activated pathways. *C. elegans* and mouse Shank proteins have been shown to promote CaV1-mediated activation of the transcription factor CREB (*Perfitt et al., 2020*; *Pym et al., 2017*).

Here, we test the idea that SHN-1 regulates CaV1 coupling to a second effector, calcium activated potassium currents (which are mediated by BK channels). BK channels are activated by both membrane depolarization and by cytoplasmic calcium. At resting cytoplasmic calcium levels (~100 nM), BK channels have extremely low open probability. Following depolarization, cytoplasmic calcium rises thereby activating BK channels. BK channels bind calcium with a $K_d$ ranging from 1 to 10 µM (*Contreras et al., 2013*); consequently, efficient BK channel activation requires tight spatial coupling to voltage activated calcium (CaV) channels. BK channels associate with all classes of CaV channels (*Berkefeld et al., 2006*). The co-clustering of BK and CaV channels allows rapid activation of hyperpolarizing potassium currents following depolarization. BK channels decrease action potential (AP) durations, promote rapid after hyperpolarization potentials, decrease the duration of calcium entry, and limit secretion of neurotransmitters and hormones in neurons and muscles (*Adams et al., 1982*; *Edgerton and Reinhart, 2003*; *Petersen and Maruyama, 1984*; *Storm, 1987*). Thus, BK channels have profound effects on circuit activity.

*C. elegans* has two BK channel subunits (SLO-1 and SLO-2), both of which form calcium and voltage dependent potassium channels when heterologously expressed (*Wang et al., 2001*; *Yuan et al., 2000*). SLO-2 channels are also activated by cytoplasmic chloride (*Yuan et al., 2000*). As in mammals, neuronal SLO-1 and –2 channels inhibit neurotransmitter release (*Liu et al., 2014*; *Liu et al., 2007*; *Sancar et al., 2011*), presumably via their coupling to UNC-2/CaV2 and EGL-19/CaV1. In body muscles, SLO-1 channels are co-localized with EGL-19/CaV1 channels and regulate muscle excitability and behavior (*Kim et al., 2009*). Here we show that SHN-1 promotes BK coupling to EGL-19/CaV1 channels, thereby decreasing AP duration.

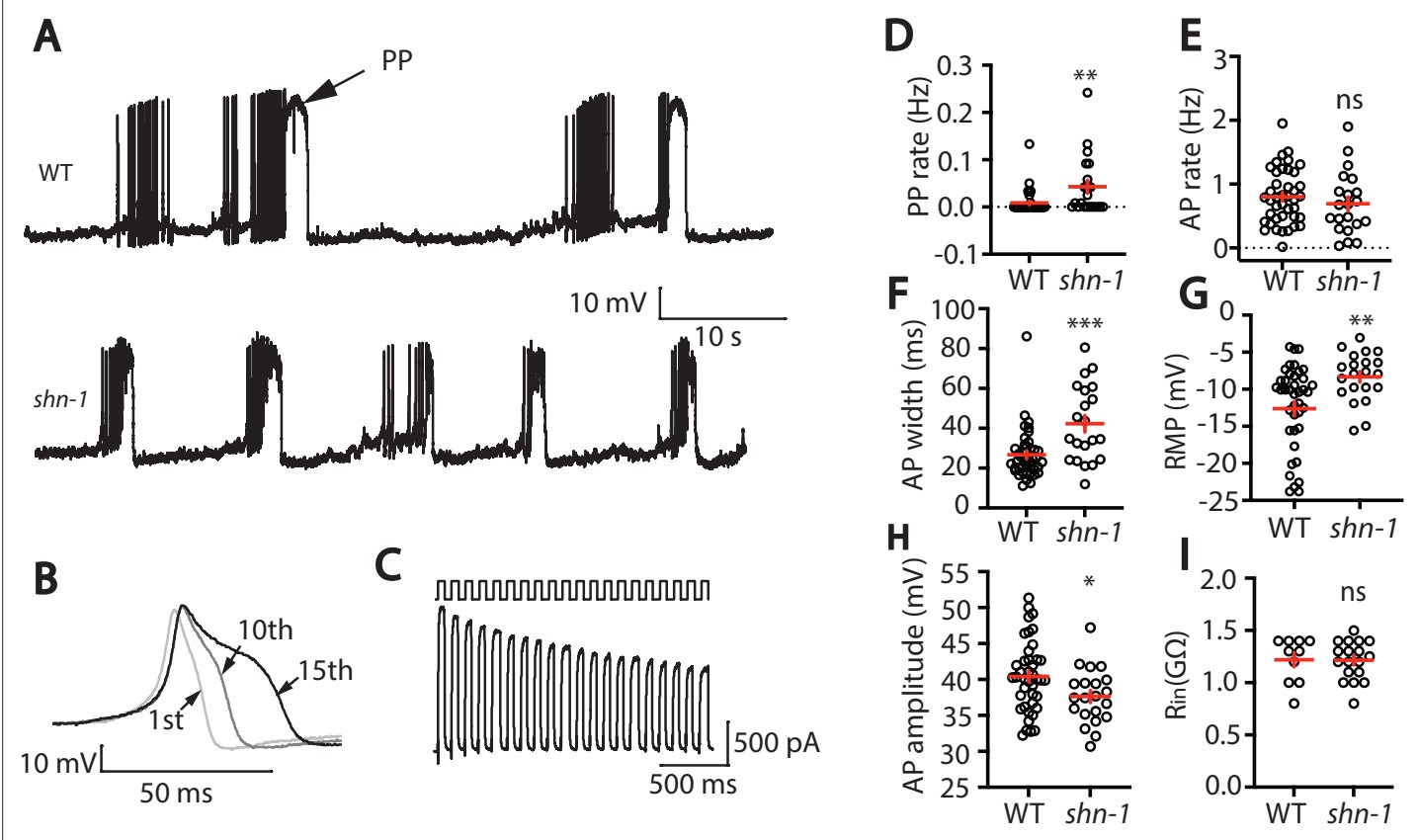

**Figure 1.** SHN-1 regulates muscle AP firing patterns. (**A**) Representative traces of spontaneous muscle APs are shown for WT and *shn-1(nu712* null) mutants. APs occur in bursts of ~10 APs/ burst. Plateau potentials (PPs), defined as transients lasting >150ms, are observed less frequently, often at the end of a burst. (**B**) APs become progressively longer during bursts. Successive APs taken from a representative burst are shown. (**C**) Repetitive depolarization to +30 mV leads to a progressive decrease in potassium currents. A representative recording from a WT animal is shown. This likely results from an accumulation of inactivated potassium channels during repetitive stimulation. (**D–I**) Mean PP rate (**D**), AP rate (**E**), AP width (**F**), RMP (**G**), AP amplitude (**H**), and input resistance ($R_{in}$, **I**) are compared in WT and *shn-1* null mutants. All *shn-1* data were obtained from *shn-1(nu712)* except for $R_{in}$ (**I**), which were from *shn-1(tm488)*. Values that differ significantly from wild type controls are indicated (ns, not significant; *, $p < 0.05$; **, $p < 0.01$; ***, $p < 0.001$). Error bars indicate SEM.

The online version of this article includes the following figure supplement(s) for figure 1:

**Figure supplement 1.** SHN-1 is expressed in many tissues.

## Results

### SHN-1 acts in muscles to regulate action potential duration

EGL-19/CaV1 channels mediate the primary depolarizing current during body muscle APs (*Jospin et al., 2002*; *Liu et al., 2011*). Because SHN-1 directly binds EGL-19 (*Pym et al., 2017*), we asked if SHN-1 regulates muscle AP firing patterns. In WT animals, body muscles exhibit a pattern of spontaneous AP bursts (~10 APs/burst; burst frequency 0.1 Hz) (*Figure 1A*). Within a burst, APs became progressively wider (*Figure 1B*). A similar pattern of progressive AP broadening during burst firing has been reported for many neurons (*Geiger and Jonas, 2000*; *Jackson et al., 1991*). Outward potassium currents were progressively decreased during repetitive depolarization (*Figure 1C*), suggesting that progressive AP broadening most likely results from accumulation of inactivated potassium channels during bursts, as seen in other cell types (*Geiger and Jonas, 2000*; *Kole et al., 2007*). Occasionally, WT muscles also exhibited prolonged depolarizations ( > 150ms), which are hereafter designated plateau potentials (PPs). PPs often occur at the end of an AP burst (*Figure 1A*).

In *shn-1* null mutants, PP rate and AP widths were significantly increased, AP amplitudes were decreased, resting membrane potential (RMP) was depolarized, while AP frequency and input resistance were unaltered (*Figure 1D–I*). Similar increases in AP widths and PP rate were observed in

**Table 1.** Comparison of *shn-1* null alleles.

| Genotype: | PP rate (Hz): | AP width (ms): | AP Amp. (mV): | RMP (mV): |
|---|---|---|---|---|
| WT | 0.01 ± 0.00 | 26.73 ± 1.98 | 40.42 ± 0.77 | −12.61 ± 0.96 |
| *shn-1(nu712)* | 0.04 ± 0.01** | 42.28 ± 4.17*** | 37.63 ± 0.83* | −8.33 ± 0.72** |
| *shn-1(nu652)* | 0.05 ± 0.02*** | 58.06 ± 5.27*** | 34.78 ± 0.92*** | −7.19 ± 0.99*** |
| *shn-1(tm488)* | 0.05 ± 0.01*** | 49.06 ± 7.07*** | 41.36 ± 1.64 | −12.06 ± 1.54 |

Mean, SEM, and significant differences from WT controls are indicated (*, $P < 0.05$; **, $P < 0.01$; ***, $P < 0.001$).

three, independently derived *shn-1* null mutants (*nu712*, *nu652*, and *tm488*) (**Table 1**). Single-cell RNA sequencing studies suggest that SHN-1 is expressed in muscles, neurons, glia, and epithelial cells (**Cao et al., 2017**; **Packer et al., 2019**), consistent with the broad expression of split GFP tagged *shn-1(nu600* GFP$_{11}$) (**Figure 1—figure supplement 1**). To determine if SHN-1 functions in body muscles to control AP duration, we edited the endogenous *shn-1* locus to construct alleles that are either inactivated (*nu697*) or rescued (*nu652*) by the CRE recombinase (**Figure 2A**). Using these alleles, we found that AP widths and PP frequency were increased in *shn-1*(muscle Knockout, KO) and that

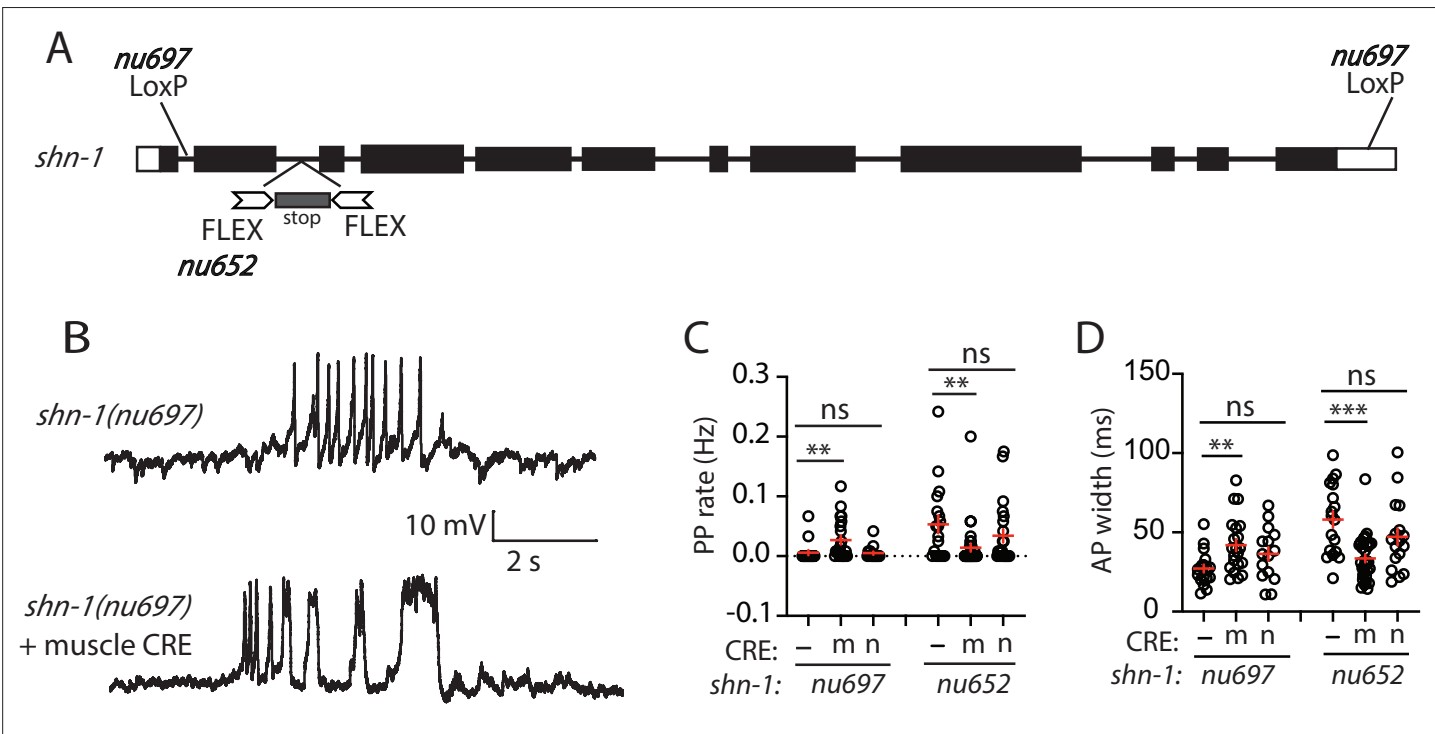

**Figure 2.** SHN-1 acts in muscles to control AP duration. (**A**) A schematic of the *shn-1* locus is shown. Open boxes indicate UTRs, black boxes indicate coding regions. Recombination sites mediating CRE induced deletions (LoxP) and inversions (FLEX) are indicated. The *shn-1(nu697)* allele allows CRE-induced *shn-1* knockouts while *shn-1(nu652)* allows CRE-induced *shn-1* rescue. In *shn-1(nu652)*, an exon containing in frame stop codons was inserted into the second intron (in the 'OFF' orientation). This stop exon is bounded by FLEX sites. (**B**) Representative traces of spontaneous muscle APs are shown in *shn-1(nu697)* with and without muscle CRE expression. Mean PP rate (**C**) and AP width (**D**) are compared in the indicated *shn-1* mutants without (-) and with CRE expression in muscles (**m**) or neurons (**n**). Sample sizes are as follows: *shn-1(nu697)* (17); *shn-1(nu697)* +muscle CRE (21); *shn-1(nu697)* +neuron CRE (15); *shn-1(nu652)* (18); *shn-1(nu652)* +muscle CRE (30); and *shn-1(nu652)* +neuron CRE (19). Values that differ significantly from wild-type controls are indicated (ns, not significant; *, p < 0.05; **, p < 0.01; ***, p < 0.001). Error bars indicate SEM. Representative traces for genotypes in panels C and D are shown in **Figure 2—figure supplement 1**.

The online version of this article includes the following figure supplement(s) for figure 2:

**Figure supplement 1.** Representative traces for recordings summarized in **Figure 2C and D**.

**Figure supplement 2.** Muscle AP defects in *shn-1(null)* and *shn-1(muscle KO)* are not significantly different.

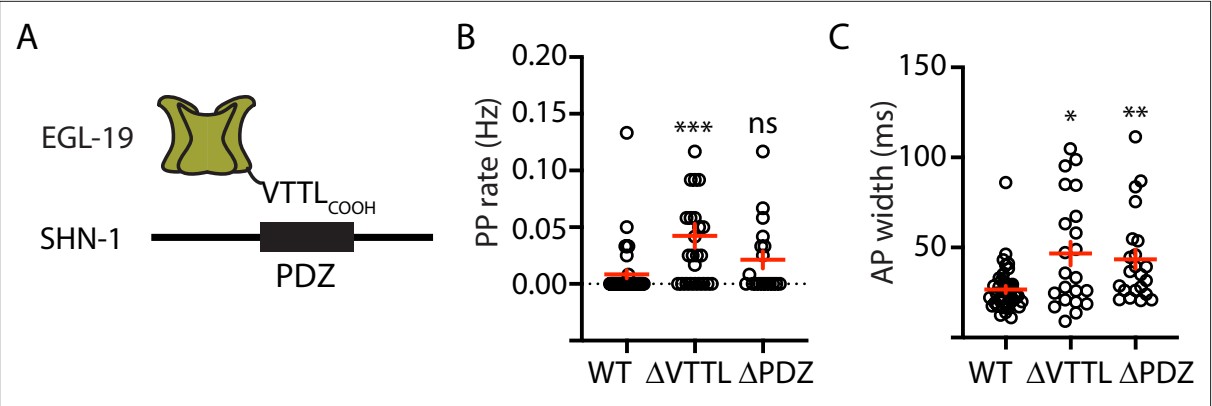

**Figure 3.** Mutations disrupting SHN-1 binding to EGL-19 increase AP duration. (**A**) A schematic illustrating the binding interaction between EGL-19's c-terminus and SHN-1's PDZ domain is shown. (**B–C**) Mean PP rate and AP width are compared in the indicated genotypes. Representative traces are shown in *Figure 3—figure supplement 1*. Mutations deleting the SHN-1 PDZ domain (*nu542 ΔPDZ*) or those deleting EGL-19's c-terminal PDZ ligand (*nu496 ΔVTTL*) were edited into the endogenous genes using CRISPR. These mutations significantly increased AP width, compared to WT controls. Sample sizes are as follows: WT (41), *shn-1(nu542)* (22), and *egl-19(nu496)* (22). Values that differ significantly from wild type controls are indicated (ns, not significant; *, p < 0.05; **, p < 0.01; ***, p < 0.001). Error bars indicate SEM.

The online version of this article includes the following figure supplement(s) for figure 3:

**Figure supplement 1.** Representative traces for recordings summarized in *Figure 3B and C*.

**Figure supplement 2.** Muscle AP defects in *shn-1(null)* and *shn-1(null); egl-19(ΔVTTL)* double mutants are not significantly different.

this defect was eliminated in *shn-1*(muscle rescue) (*Figure 2B–D*). By contrast, *shn-1*(neuron KO) and *shn-1*(neuron rescue) had no effect on PP rate or AP widths (*Figure 2C–D*). Because CRE expression in muscles produced opposite changes in AP firing patterns in strains containing the *shn-1 nu697* and *nu652* alleles, these results are unlikely to be caused by toxicity associated with CRE expression (*Speed et al., 2019*). The PP rate and AP width defects observed in *shn-1*(muscle KO) were not significantly different from those in *shn-1(null)* (*Figure 2—figure supplement 2*). Collectively, these results suggest that SHN-1 acts in body muscles to control AP duration.

## SHN-1 binding to EGL-19 promotes AP repolarization

Because SHN-1 has multiple binding partners, we sought to confirm that prolonged APs result from decreased SHN-1 binding to EGL-19. To address this question, we recorded APs in strains containing mutations that disrupt this interaction (*Figure 3A*). PP frequency was significantly increased by a deletion removing the EGL-19 carboxy-terminal PDZ ligand [*egl-19(nu496 ΔVTTL)*] (*Figure 3B*). AP widths were significantly increased by a deletion removing the SHN-1 PDZ domain [*shn-1(nu542 ΔPDZ)*] and by the *egl-19(nu496 ΔVTTL)* mutation (*Figure 3C*). Furthermore, the *shn-1(nu712 null)* and *egl-19(nu496 ΔVTTL)* mutations did not have additive effects on PP rate and AP widths in double mutants (*Figure 3—figure supplement 2*). Taken together, these results support the idea that SHN-1 binding to EGL-19/CaV1 accelerates AP repolarization.

## AP repolarization is controlled by SHK-1 KCNA and SLO-1/2 BK channels

To investigate how SHN-1 controls AP duration, we first asked which potassium channels promote repolarization following APs. Prior studies showed that voltage-activated potassium currents in body muscles are mediated by SHK-1/KCNA and BK channels (*Gao and Zhen, 2011*; *Liu et al., 2011*). SHK-1 channel function can be assessed in recordings using an internal solution containing low chloride levels (hereafter Ik$_{loCl}$). Ik$_{loCl}$ was nearly eliminated in *shk-1* single mutants (*Figure 4A–B*). BK channel function can be assayed in recordings utilizing internal solutions with high chloride levels (hereafter Ik$_{hiCl}$), which activates SLO-2 channels (*Yuan et al., 2000*). Ik$_{hiCl}$ was ~50% reduced in single mutants lacking either SLO-2 or SHK-1 and was eliminated in *slo-2; shk-1* double mutants (*Figure 4C–D*). These results suggest that SHK-1/KCNA and SLO-2/BK are the primary channels promoting AP repolarization.

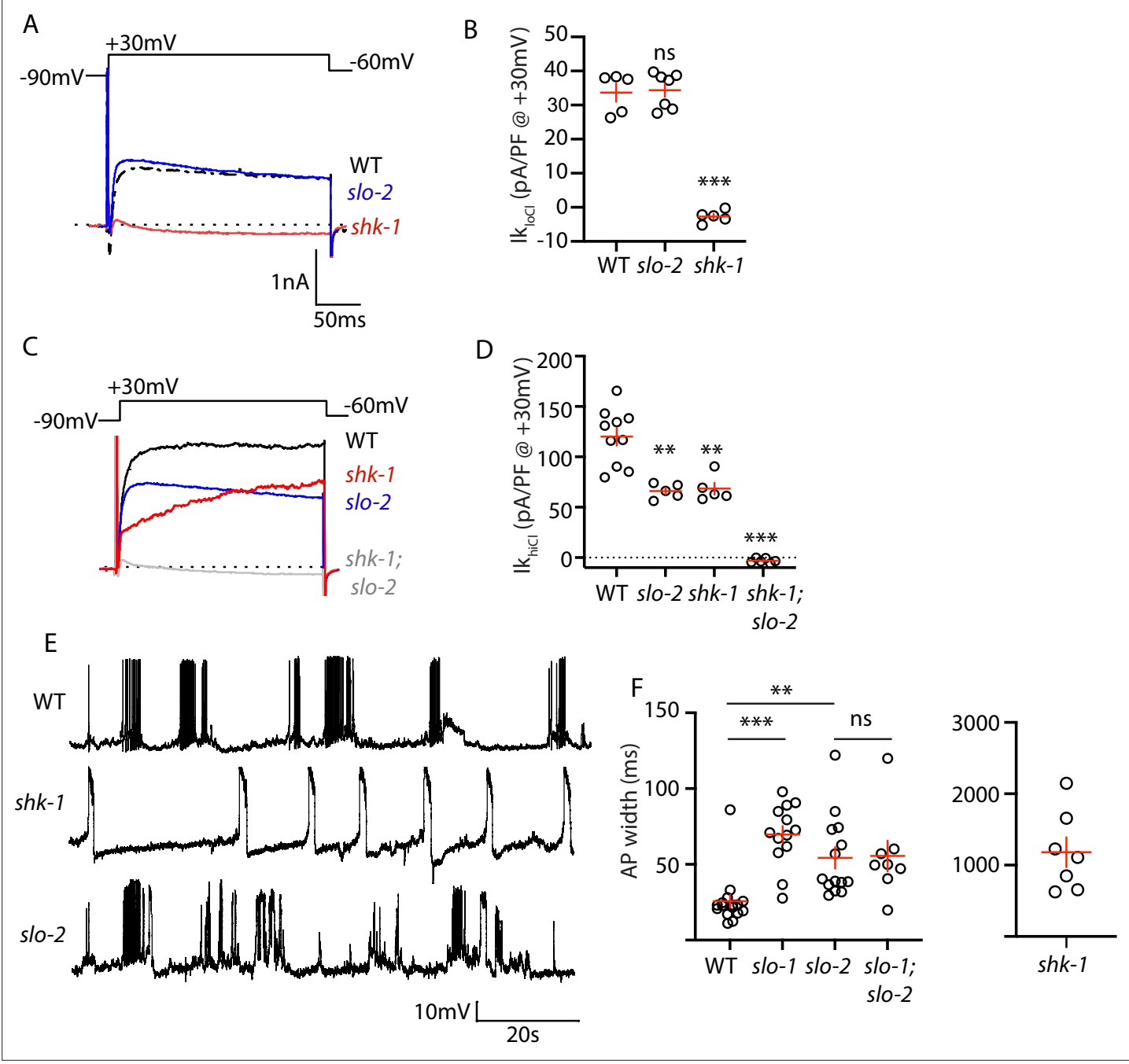

**Figure 4.** AP repolarization is mediated by SHK-1 and SLO channels. (**A–D**) Muscle voltage activated potassium currents are mediated by SHK-1 and SLO-2. Voltage activated potassium currents were recorded using pipette solutions containing low (Ik$_{loCl}$, **A–B**) and high (Ik$_{hiCl}$, **C–D**) chloride concentrations. Representative traces (**A,C**) and mean current density (**B,D**) at +30 mV are shown. Ik$_{loCl}$ is mediated by SHK-1 whereas SHK-1 and SLO-2 equally contribute to Ik$_{hiCl}$. (**E–F**) AP durations are significantly increased in mutants lacking SHK-1, SLO-1, and SLO-2 channels. The AP widths observed in *slo-1; slo-2* double mutants were not significantly different from those found in *slo-2* single mutants. Representative traces (E and *Figure 4—figure supplement 1*) and mean AP widths (**F**) are shown. Alleles used in this figure were: *shk-1(ok1581)*, *slo-1(js379)*, and *slo-2(nf100)*. Sample sizes are as follows: in panel B, WT (5), *slo-2* (7), and *shk-1* (5); in panel D, WT (10), *slo-2* (5), *shk-1* (5), *shk-1;slo-2* (6); in panel F, WT (16), *slo-1* (13), *slo-2* (14), *slo-1; slo-2* (8), *shk-1* (7). Values that differ significantly from wild type controls are indicated (ns, not significant; *, p < 0.05; **, p < 0.01; ***, p < 0.001). Error bars indicate SEM.

The online version of this article includes the following figure supplement(s) for figure 4:

**Figure supplement 1.** Representative traces for recordings summarized in *Figure 4E*.

Consistent with this idea, AP duration was significantly increased in mutants lacking SHK-1/KCNA (*Figure 4E–F*), as previously reported (*Gao and Zhen, 2011*; *Liu et al., 2011*).

Contradictory results have been reported for AP firing patterns in *slo-1* and *slo-2* BK mutants (*Gao and Zhen, 2011*; *Liu et al., 2011*). These studies used intracellular solutions that alter BK channel function. In (*Liu et al., 2011*), an intracellular solution containing high chloride levels was used, thereby exaggerating SLO-2's contribution to AP repolarization (*Yuan et al., 2000*). In (*Gao and Zhen, 2011*), an intracellular solution containing a fast calcium chelator (BAPTA) was used, which inhibits BK activation thereby minimizing their impact on APs. We re-investigated the effect of SLO channels on APs using intracellular solutions with low chloride and a slow calcium chelator (EGTA), finding that AP durations were increased to a similar extent in *slo-1* and *slo-2* single mutants (*Figure 4E–F*). Taken together, these results confirm that SHK-1/KCNA and SLO/BK are the primary channels promoting AP repolarization in body muscles.

## SLO-1 and SLO-2 function together to promote AP repolarization

SLO-1 and SLO-2 subunits are co-expressed in muscles and could potentially co-assemble to form heteromeric channels. To determine if channels containing both SLO-1 and SLO-2 regulate AP repolarization, we analyzed AP widths in *slo-1; slo-2* double mutants. AP widths in *slo-1; slo-2* double mutants were not significantly different from those found in the single mutants (*Figure 4F*). Because *slo-1* and *slo-2* mutations did not have additive effects on AP widths, these results support the idea that heteromeric SLO-1/2 channels mediate rapid repolarization of muscle APs.

We did several experiments to further test the idea that SLO-1 and SLO-2 function together in heteromeric channels. First, we recorded voltage-activated potassium current in body muscles and found that $Ik_{hiCl}$ was modestly reduced in *slo-1* mutants, was dramatically reduced in *slo-2* mutants, and was not further reduced in *slo-1; slo-2* double mutants (*Figure 5A–B*). These results suggest that $Ik_{hiCl}$ is mediated by heteromeric channels (containing both SLO-1 and SLO-2 subunits) and by SLO-2 homomers.

As a final test of this idea, we asked if subcellular localization of SLO-1 and SLO-2 subunits requires expression of both subunits. For this analysis, endogenous SLO-1 and SLO-2 subunits were labeled with split GFP. Using CRISPR, we introduced the eleventh β-strand of GFP ($GFP_{11}$) into the endogenous *slo-1* and *slo-2* genes and visualized their expression by expressing $GFP_{1-10}$ in body muscles. Strains containing the $GFP_{11}$ tagged alleles exhibited wild-type AP widths, RMP, and $Ik_{hiCl}$ currents, indicating that the tag did not interfere with SLO channel function (*Figure 5—figure supplement 1*). Using these alleles, we find that SLO-2 puncta intensity was significantly reduced in *slo-1* null mutants, indicating that channels containing SLO-2 subunits require SLO-1 for their trafficking (*Figure 5C–D*). By contrast, SLO-1 puncta intensity was unaffected in *slo-2* mutants, suggesting that BK channels lacking SLO-2 were trafficked normally (*Figure 5E–F*). Collectively, these results suggest that rapid muscle repolarization following APs is mediated by SLO-1/2 heteromeric channels and by SLO-2 homomers. Two prior studies also suggested that SLO subunits form heteromeric channels when heterologously expressed in *Xenopus* oocytes. SLO-1 currents were inhibited by a dominant-negative SLO-2 construct (*Yuan et al., 2000*). Similarly, mammalian SLO2 subunits (KCNT1 and 2) co-assemble to form heteromeric channels (*Chen et al., 2009*). Our results suggest that endogenously expressed SLO subunits also form heteromeric channels in native tissues.

## SHN-1 controls AP duration through BK channels

SHN-1's impact on AP duration could be mediated by changes in either SHK-1 or SLO channels. To determine if SHN-1 acts through SLO channels, we asked if *slo-2* mutations block SHN-1's effects on AP widths. Consistent with this idea, AP widths in *slo-2* single mutants were not significantly different from those in *slo-2* double mutants containing *shn-1(nu712 null)*, *shn-1(nu542 ΔPDZ)*, or *egl-19(nu496 ΔVTTL)* mutations (*Figure 6A*). These results suggest that SHN-1 controls AP duration by regulating SLO-2 channels.

To confirm that SHN-1 regulates SLO-2 channels, we analyzed potassium currents in *shn-1* mutants. A *shn-1* null mutation had no effect on $Ik_{loCl}$ currents, indicating that SHK-1 function was unaffected (*Figure 6B–C*). By contrast, $Ik_{hiCl}$ was ~30% reduced in *shn-1* null mutants, ~ 50% reduced in *slo-2* mutants, and was not further reduced in *shn-1; slo-2* double mutants (*Figure 6D–E*). Lack of additivity in *shn-1; slo-2* double mutants suggests that the SHN-1-sensitive potassium current was mediated by

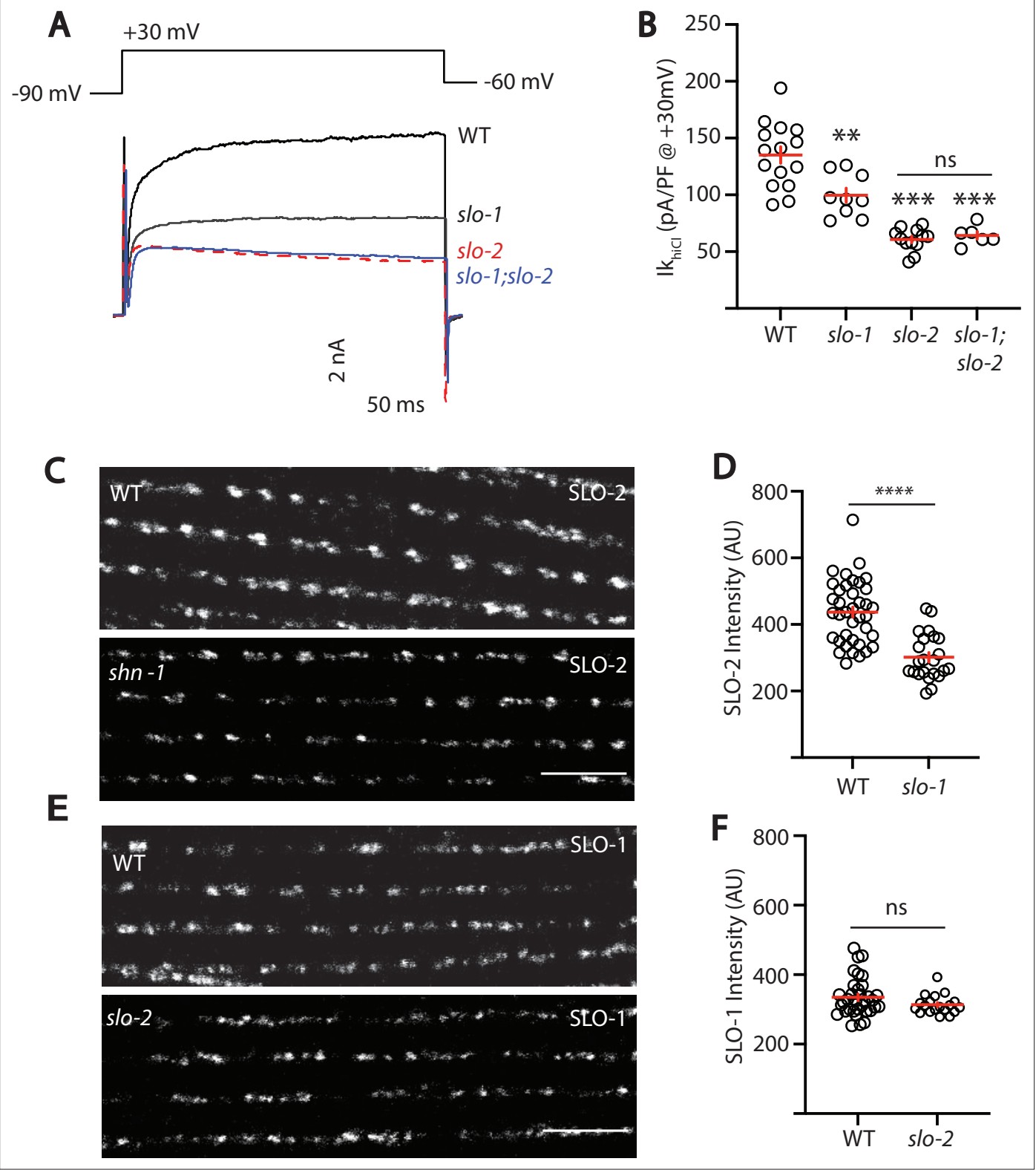

**Figure 5.** SLO-2 and SLO-1 function together in heteromeric channels. (**A–B**) $Ik_{hiCl}$ was significantly decreased in *slo-1(js379)* and *slo-2(nf100)* single mutants but was not further decreased in *slo-1; slo-2* double mutants. Representative traces (**A**) and mean current density (**B**) at +30 mV are shown. Sample sizes for panel B: WT (15), *slo-1* (9), *slo-2* (12), *slo-1;slo-2* (6). (**C–F**) Expression of split GFP tagged SLO-2 (**C–D**) and SLO-1 (**E–F**) was analyzed in body muscles. CRISPR alleles were constructed adding 7 copies of $GFP_{11}$ to the endogenous *slo-1* and *slo-2* genes (**Table 2**) and fluorescence was reconstituted by expressing $GFP_{1-10}$ in body muscles. Controls showing that the $GFP_{11}$ tags had no effect on AP width, RMP, and potassium currents are

*Figure 5 continued on next page*

Figure 5 continued

shown in **Figure 5—figure supplement 1**. Representative images (**C and E**) and mean puncta intensity (**D and F**) are shown. SLO-2 puncta intensity was significantly decreased in *slo-1(js379)* mutants. SLO-1 puncta intensity was unaltered in *slo-2(nf100)* mutants. Sample sizes are as follows: in panel D, WT (38) and *slo-1* (23); in panel F, WT (34) and *slo-2* (19). Values that differ significantly from wild type controls are indicated (ns, not significant; *, p < 0.05; **, p < 0.01; ***, p < 0.001). Error bars indicate SEM. Scale bar indicates 4 μm.

The online version of this article includes the following figure supplement(s) for figure 5:

**Figure supplement 1.** Analysis of GFP$_{11}$ tagged *slo-1* and *slo-2* alleles.

SLO-2. A similar decrease in Ik$_{hiCl}$ current was observed in *shn-1(nu542 ΔPDZ)* mutants (**Figure 6E**). Ik$_{hiCl}$ current was unaltered in *egl-19(nu496 ΔVTTL)* mutants, implying that SHN-1 binding to EGL-19's carboxy terminus is not required for SLO-2 current (**Figure 6E**). Thus, *shn-1* inactivation decreased SLO-1/2 BK current but had little or no effect on SHK-1 KCNA current; consequently, SHN-1 regulates AP widths by promoting activation of SLO-1/2 channels.

## SHN-1 promotes microdomain coupling of SLO-2 with EGL-19/CaV1 channels

BK channels bind calcium with affinities ranging from 1 to 10 μM (**Contreras et al., 2013**). As a result of this calcium dependence, BK channels have very low open probability at resting cytoplasmic calcium levels (~100 nM) and efficient BK activation typically requires close spatial coupling to calcium channels (**Barrett et al., 1982**). We next asked if body muscle BK channels are functionally coupled to EGL-19/CaV1 channels. Consistent with this idea, Ik$_{hiCl}$ current was significantly decreased by nemadipine, an EGL-19/CaV1 antagonist (**Figure 7A–B**; **Kwok et al., 2008**). The inhibitory effect of nemadipine on Ik$_{hiCl}$ was eliminated in *slo-2* mutants (**Figure 7A–B**), suggesting that the nemadipine-sensitive potassium current was mediated by SLO-2.

Is EGL-19 coupling to SLO-2 mediated by microdomain signaling? To test this idea, we compared Ik$_{hiCl}$ and AP widths recorded with intracellular solutions containing fast (BAPTA) and slow (EGTA) calcium chelators (**Figure 7C–E**). We found that Ik$_{hiCl}$ recorded with BAPTA was significantly smaller than that recorded with EGTA (**Figure 7C**). Similarly, AP widths recorded with BAPTA were significantly longer than those recorded with EGTA (**Figure 7D–E**). The effect of BAPTA on Ik$_{hiCl}$ and AP widths was eliminated in *slo-2* mutants (**Figure 7C and E**), suggesting that the BAPTA sensitive potassium current is mediated by SLO-2. BAPTA's effect on Ik$_{hiCl}$ and AP widths was reduced but not eliminated in *shn-1* mutants (**Figure 7C and E**), consistent with the partial loss of SLO-2 current in these mutants (**Figure 6D–E**). Taken together, these results suggest that SHN-1 promotes SLO-2 microdomain coupling to EGL-19/CaV1.

If BK channels are functionally coupled to EGL-19/CaV1, these channels should be co-localized. Endogenous SLO-2 channels (tagged with GFP$_{11}$) were distributed in a punctate pattern on the muscle surface. A subset of the SLO-2 puncta co-localized with EGL-19/CaV1 channels (tagged with Cherry$_{11}$), suggesting that EGL-19 nanocomplexes are heterogeneous (**Figure 7F**). SLO-2 puncta intensity was significantly reduced in *shn-1* null mutants (**Figure 7F–G**), consistent with the decreased SLO-2 current observed in these mutants. By contrast, SLO-2 puncta intensity was unaltered in *shn-1(nu542 ΔPDZ)* and *egl-19(nu496 ΔVTTL)* mutants (**Figure 7G**), in which SHN-1 binding to EGL-19's c-terminus is disrupted (**Pym et al., 2017**). Next, we asked if inactivating SHN-1 alters the localization of other muscle ion channels. SLO-1 puncta intensity was unaltered in *shn-1* null mutants, indicating that BK channels lacking SLO-2 were trafficked normally (**Figure 7—figure supplement 2A-B**). In body muscles, EGL-19/CaV1 channels are extensively co-localized with calcium channels in the endoplasmic reticulum (ER), UNC-68/Ryanodine Receptors (RYR) (**Piggott et al., 2021**). However, the puncta intensity of endogenous EGL-19(Cherry$_{11}$) and UNC-68(GFP$_{11}$)/RYR in body muscles were unaltered in *shn-1*(null) mutants (**Figure 7H** and **Figure 7—figure supplement 2C-D**), suggesting that SHN-1 does not broadly regulate co-localization of ion channels at ER-plasma membrane junctional contacts. Collectively, these results suggest that SHN-1 stabilizes SLO-2 clusters in the plasma membrane and promotes activation of heteromeric SLO-1/2 channels by nearby calcium channels.

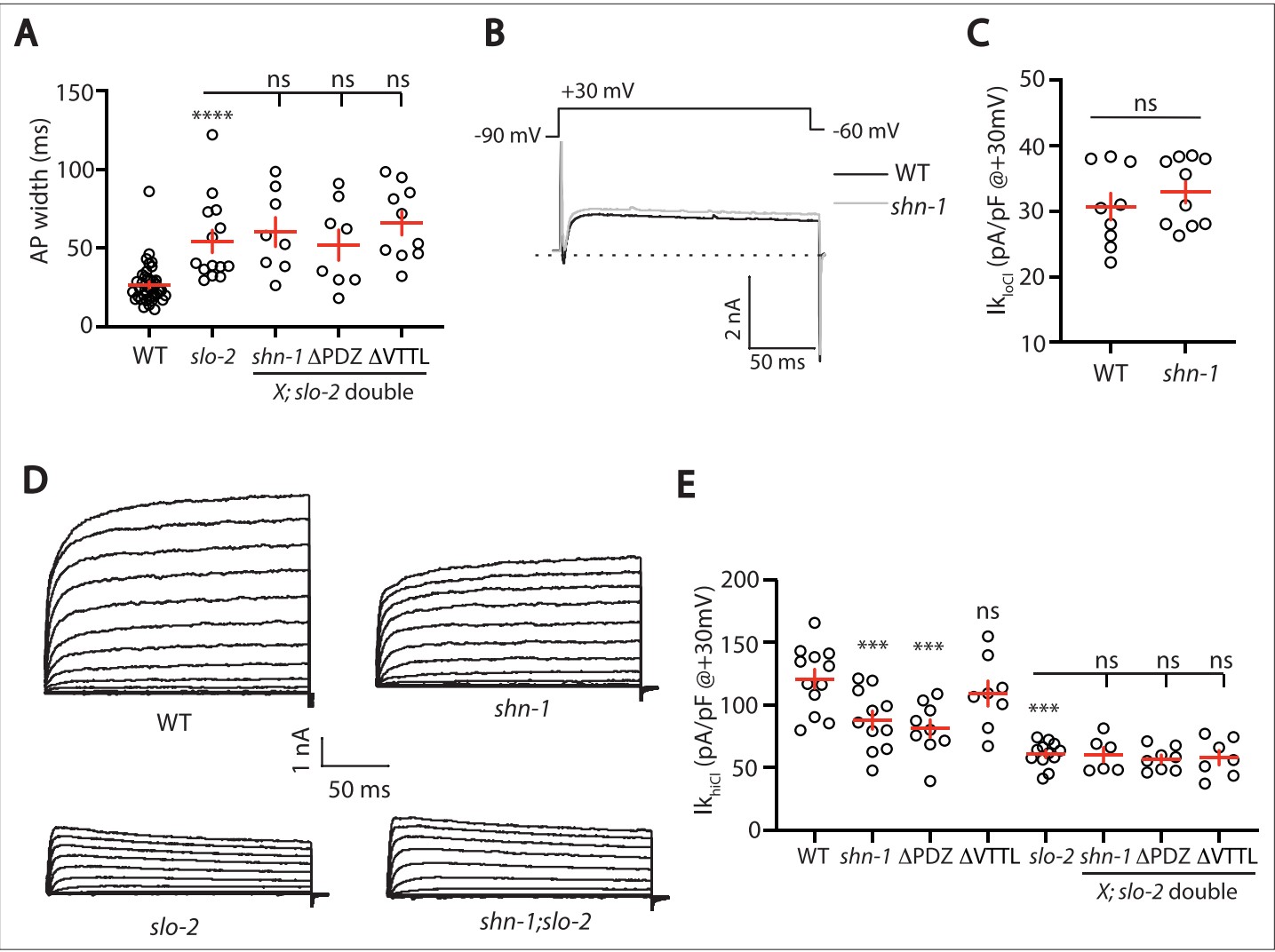

**Figure 6.** SHN-1 controls AP width by regulating SLO-2. (**A**) A *slo-2* null mutation blocks the effect of SHN-1 on AP width. Mean AP widths in *slo-2(nf100)* double mutants containing *shn-1(nu712 null)*, *shn-1(nu542 ΔPDZ)*, and *egl-19(nu496 ΔVTTL)* mutations were not significantly different from those in *slo-2* single mutants. Representative traces are shown in *Figure 6—figure supplement 1A*. Sample sizes: WT (41), *slo-2* (14), *slo-2;shn-1* (8), *slo-2;ΔPDZ* (8), and *slo-2;ΔVTTL* (10). (**B–C**) $Ik_{loCl}$ currents were unaltered in *shn-1(nu712 null)* mutants. Representative traces (**B**) and mean current density at +30 mV (**C**) are shown. Sample sizes: WT (9) and *shn-1* (10). These results show that SHK-1 channel function was unaffected in *shn-1* mutants. (**D–E**) $Ik_{hiCl}$ currents were significantly smaller in *shn-1(nu712 null)* and *shn-1(nu542 ΔPDZ)* mutants but were unaffected in *egl-19(nu496 ΔVTTL)* mutants. The effect of *shn-1* mutations on $Ik_{hiCl}$ was eliminated in double mutants lacking SLO-2, indicating that the SHN-1 sensitive potassium current is mediated by SLO-2. $Ik_{hiCl}$ currents were recorded from adult body wall muscles of the indicated genotypes at holding potentials of –60 to +60 mV. Representative traces (D and *Figure 6—figure supplement 1B*) and mean current density at +30 mV (**E**) are shown. Sample sizes in panel E: WT (12), *shn-1* (11), *slo-2* (12), *ΔPDZ* (9), *ΔVTTL* (8), *slo-2;shn-1* (6), *slo-2;ΔPDZ* (8), and *slo-2;ΔVTTL* (7). Values that differ significantly from wild type controls are indicated (ns, not significant; *, p < 0.05; **, p < 0.01; ***, p < 0.001). Error bars indicate SEM.

The online version of this article includes the following figure supplement(s) for figure 6:

**Figure supplement 1.** Representative traces for recordings summarized in *Figure 6A and E*.

## EGL-19 to SLO-2 coupling is sensitive to *shn-1* gene dose

Deletion and duplication of human shank genes are both associated with ASD, schizophrenia, and mania (*Bonaglia et al., 2006*; *Durand et al., 2007*; *Failla et al., 2007*; *Gauthier et al., 2010*; *Han et al., 2013*). These results suggest that Shank phenotypes relevant to psychiatric disorders should exhibit a similar sensitivity to Shank copy number. For this reason, we analyzed the effect of *shn-1* gene dosage on $Ik_{hiCl}$ and AP duration (*Figure 8*). We analyzed animals with 1 (*nu712/+* heterozygotes), 2 (WT), and 4 (WT +2 single copy *shn-1* transgenes) copies of *shn-1*. Compared to wild-type

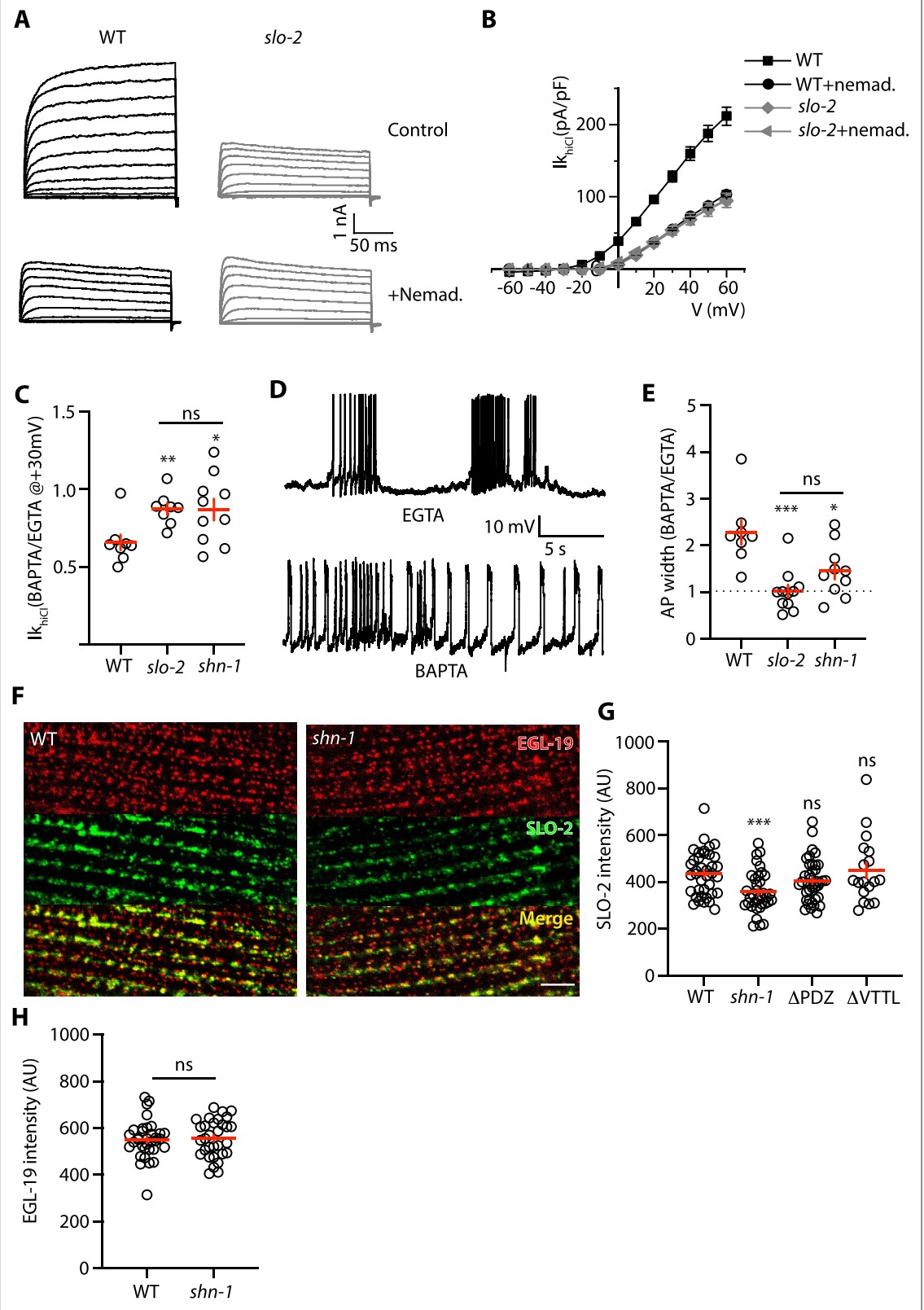

**Figure 7.** SHN-1 promotes EGL-19 to SLO-2 microdomain coupling. (**A–B**) SLO-2 activation is functionally coupled to EGL-19. $Ik_{hiCl}$ was significantly reduced by nemadipine (an EGL-19 antagonist). This inhibitory effect of nemadipine on $Ik_{hiCl}$ was eliminated in *slo-2(nf100)* mutants, indicating that the nemadipine sensitive current is mediated by SLO-2. $Ik_{hiCl}$ currents were recorded from adult body wall muscles of the indicated genotypes at holding potentials of –60 to +60 mV. Representative $Ik_{hiCl}$ traces (**A**) and mean current density as a function of membrane potential (**B**) are shown. (**C**) SLO-2

*Figure 7 continued on next page*

*Figure 7 continued*

activation requires microdomain coupling to EGL-19. $Ik_{hiCl}$ currents recorded in BAPTA are significantly smaller than those in EGTA. The inhibitory effect of BAPTA was reduced in *shn-1(nu712* null) mutants and was eliminated in *slo-2(nf100)* mutants, indicating that the BAPTA sensitive current is mediated by SLO-2. The ratio of $Ik_{hiCl}$ current density at +30 mV recorded in BAPTA to the mean current density recorded in EGTA is plotted for the indicated genotypes. Representative traces are shown in *Figure 7—figure supplement 1A*. Sample sizes for panel C: WT (8), *slo-2* (8), and *shn-1* (10). (**D–E**) AP repolarization is mediated by microdomain activation of SLO-2. AP widths recorded in solutions containing BAPTA are wider than those recorded in EGTA. The effect of BAPTA on AP widths was reduced in *shn-1(nu712* null) mutants and was eliminated in *slo-2(nf100)* mutants, indicating that BAPTA's effect is mediated by SLO-2. Representative traces of WT muscle APs recorded in EGTA and BAPTA are shown (**D**). The ratio of AP widths recorded in BAPTA to the mean AP widths recorded in EGTA is plotted for the indicated genotypes (**E**). Representative traces for panel E are shown in *Figure 7—figure supplement 1B*. Sample sizes for panel E: WT (8), *slo-2* (11), and *shn-1* (10). (**F–H**) SLO-2(nu725 $GFP_{11}$) is partially co-localized with EGL-19(nu722 $Cherry_{11}$) in body muscles. $GFP_{11}$ and $Cherry_{11}$ fluorescence were reconstituted by expressing $GFP_{1-10}$ and $Cherry_{1-10}$ in body muscles. SLO-2 puncta intensity was significantly reduced in *shn-1(nu712* null) mutants but was unaffected in *shn-1(nu542* ΔPDZ) and *egl-19(nu496* ΔVTTL) mutants. Representative images (**F**) and mean puncta intensity for SLO-2 (**G**) and EGL-19 (**H**) are shown. Sample sizes for panel G: *slo-2*($GFP_{11}$) single mutants (38), and double mutants containing the *shn-1* (35), ΔPDZ (39), and ΔVTTL (18) mutations. Sample sizes for panel H: *egl-19*($Cherry_{11}$) single mutants (34) and *shn-1; egl-19*($Cherry_{11}$) double mutants (31). Values that differ significantly from wild type controls are indicated (ns, not significant; *, $p < 0.05$; **, $p < 0.01$; ***, $p < 0.001$). Error bars indicate SEM. Scale bar indicates 4 μm.

The online version of this article includes the following figure supplement(s) for figure 7:

**Figure supplement 1.** Representative traces for recordings summarized in *Figure 7C and E*.

**Figure supplement 2.** SLO-1 and UNC-68/RYR puncta intensity is unaltered in *shn-1* mutant muscles.

controls, AP duration was significantly increased (*Figure 8A and B*) while muscle $Ik_{hiCl}$ was significantly decreased (*Figure 8C and D*) in animals containing 1 and 4 copies of *shn-1*. Thus, increased and decreased *shn-1* gene dosage produced similar defects in AP duration and SLO-2 current.

### SHN-1 regulates BK channel activation in motor neurons

Thus far, our results suggest that SHN-1 promotes EGL-19 to SLO-2 coupling in muscles. We next asked if SHN-1 also promotes coupling in motor neurons. To test this idea, we analyzed $Ik_{hiCl}$ in cholinergic motor neurons and found that it was significantly reduced in *shn-1* null mutants (*Figure 9A–C*). The *slo-2* and *shn-1* mutations did not have additive effects on $Ik_{hiCl}$ in double mutants, suggesting that the *shn-1* mutation selectively decreases SLO-2 current in motor neurons (*Figure 9A–C*). Consistent with decreased SLO-2 currents, we observed a corresponding decrease in axonal SLO-2(nu725 $GFP_{11}$) puncta fluorescence in *shn-1* mutant motor neurons (*Figure 9D–E*). Thus, our results suggest that SHN-1 promotes EGL-19/CaV1 to SLO-2 coupling in both body muscles and cholinergic motor neurons.

## Discussion

Our results lead to six principal conclusions. First, we show that SHN-1 acts cell autonomously in muscles to promote rapid repolarization of APs. Second, heteromeric BK channels containing both SLO-1 and SLO-2 subunits promote AP repolarization. Third, SHN-1 limits AP duration by promoting BK channel activation. Fourth, *shn-1* mutants have decreased SLO-2 channel clustering and decreased SLO-2 currents. Fifth, increased and decreased SHN-1 gene dosage produce similar defects in AP durations and SLO-2 currents. And sixth, SHN-1 also promotes SLO-2 activation in motor neurons. Below we discuss the significance of these findings.

### Shank as a regulator of ion channel density

Several recent studies suggest that an important function of Shank proteins is to regulate ion channel density and localization. Mutations inactivating Shank have been shown to decrease AMPA and NMDA receptor abundance and post-synaptic currents (*Peça et al., 2011*; *Won et al., 2012*), HCN channels (*Yi et al., 2016*; *Zhu et al., 2018*), TRPV channels (*Han et al., 2016*), and voltage-activated CaV1 calcium channels (*Pym et al., 2017*; *Wang et al., 2017*). Here, we show that Shank also regulates BK channel densities in *C. elegans* muscles and motor neurons. Collectively, these studies suggest that Shank proteins have the capacity to control localization of many ion channels, thereby shaping neuron and muscle excitability.

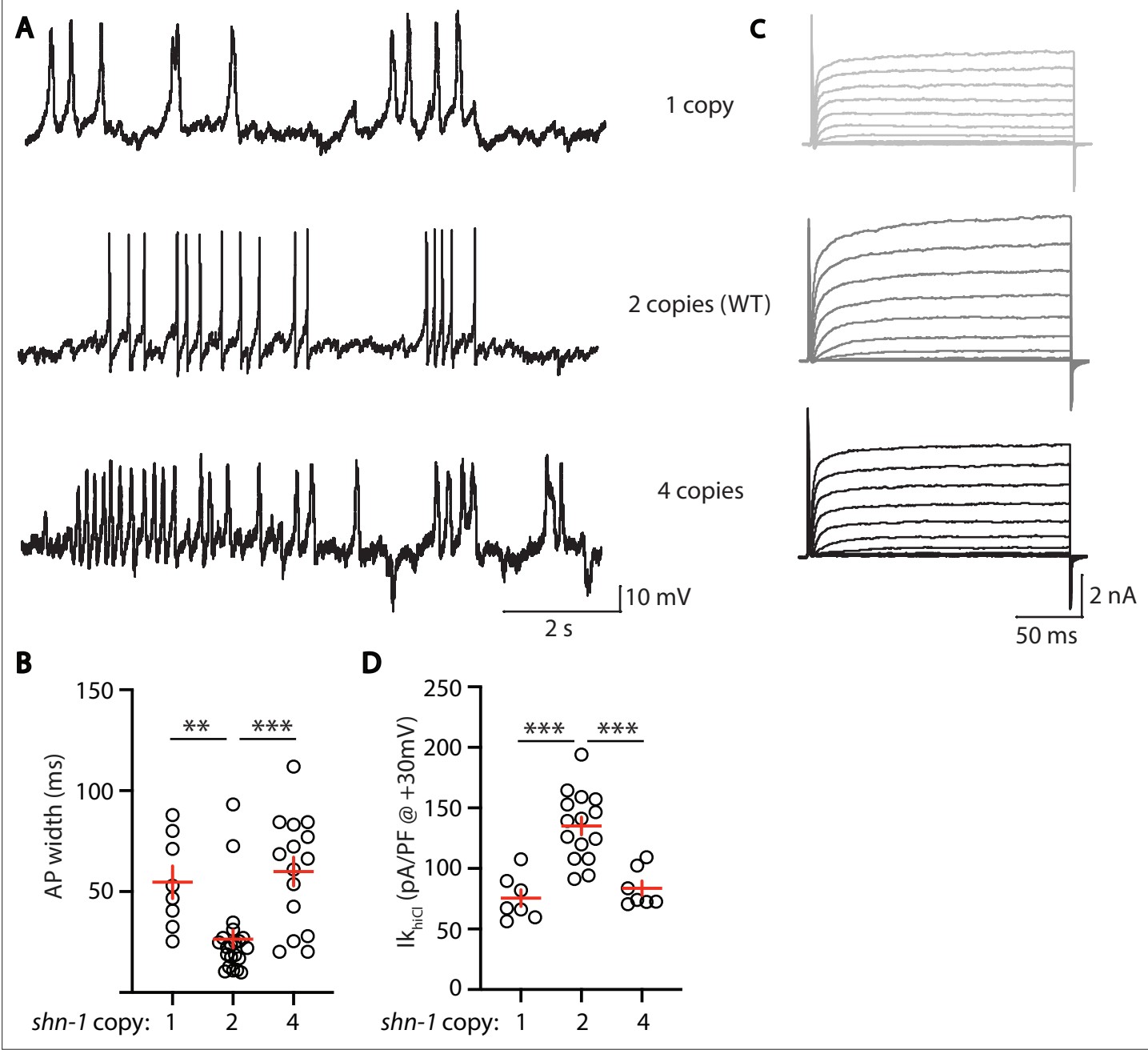

**Figure 8.** AP width and SLO-2 current are sensitive to *shn-1* gene dosage. The effect of *shn-1* gene dosage on AP widths (**A–B**) and Ik$_{hiCl}$ current (**C–D**) was analyzed. Ik$_{hiCl}$ was significantly decreased while AP duration was significantly increased in animals containing 1 and 4 copies of *shn-1* compared to WT controls (i.e. 2 copies). The following genotypes were analyzed: 1 copy of *shn-1* [*shn-1(nu712)/ +* heterozygotes], 2 copies of *shn-1* (WT) and 4 copies of *shn-1* (*nuSi26* homozygotes in wild-type). Ik$_{hiCl}$ currents were recorded from adult body wall muscles of the indicated genotypes at holding potentials of –60 to +60 mV. Representative traces (**A,C**), mean AP width (**B**), and mean Ik$_{hiCl}$ current density at +30 mV (**D**) are shown. Sample sizes: for panel B, 1 copy (8), 2 copies (21), and four copies (15); for panel D, 1 copy (8), 2 copies (15), and four copies (7). Significant differences are indicated (ns, not significant; *, p < 0.05; **, p < 0.01; ***, p < 0.001). Error bars indicate SEM.

Shank regulation of BK channels could have broad effects on neuron and muscle function. In neurons, BK channels are functionally coupled to CaV channels in the soma and dendrites, thereby regulating AP firing patterns and somatodendritic calcium transients (*Golding et al., 1999*; *Storm, 1987*). In pre-synaptic terminals, BK channels limit the duration of calcium influx during APs, thereby decreasing neurotransmitter release (*Griguoli et al., 2016*; *Yazejian et al., 2000*). In muscles, BK channels regulate AP firing patterns, calcium influx during APs, and muscle contraction (*Dopico et al.,*

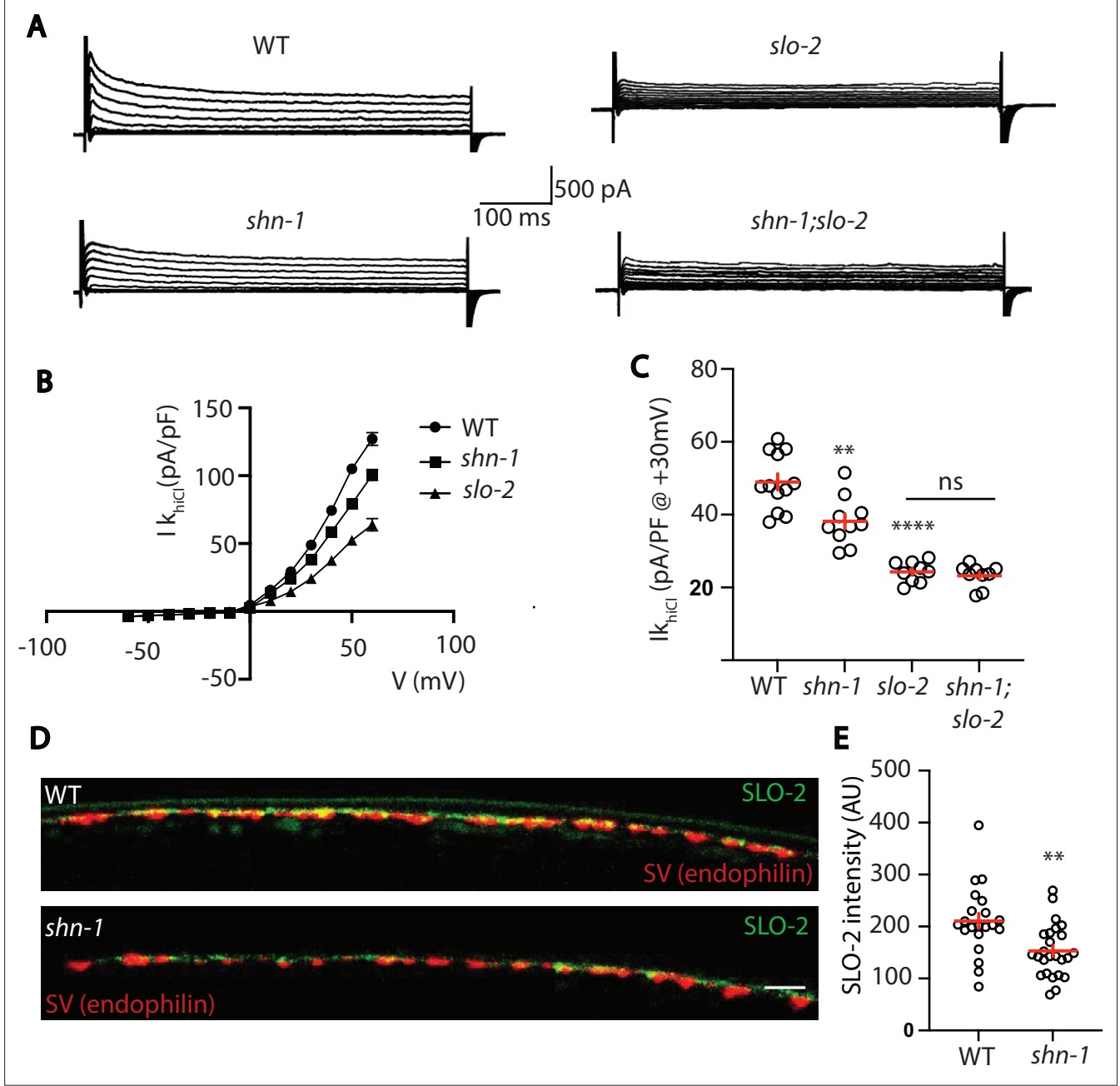

**Figure 9.** SHN-1 controls SLO-2 currents in motor neurons. (**A–B**) $Ik_{hiCl}$ currents in cholinergic motor neurons were significantly decreased in *shn-1(nu712* null) mutants. $Ik_{hiCl}$ currents were recorded from adult cholinergic motor neurons of the indicated genotypes at holding potentials of –60 to +60 mV. Representative traces (**A**), mean current density as a function of membrane potential (**B**), and mean current density at +30 mV (**C**) are shown. Sample sizes for panels B and C: WT (12), *shn-1* (10), *slo-2* (9), and *shn-1; slo-2* (9). (**D–E**) SLO-2 puncta intensity in motor neuron axons was significantly decreased in *shn-1(nu712* null) mutants. Representative images of SLO-2(*nu725* GFP₁₁) and a synaptic vesicle marker [UNC-57/Endophilin(mCherry)] in dorsal cord axons of DA/DB motor neurons are shown (**D**). GFP₁₁ fluorescence was reconstituted with GFP₁₋₁₀ expressed in DA/DB motor neurons (using the *unc-129* promoter). Mean SLO-2 puncta intensity in axons is plotted (**E**). Sample sizes for panel E: WT (21) and *shn-1* (25). Values that differ significantly from wild type controls are indicated (ns, not significant; *, $p < 0.05$; **, $p < 0.01$; ***, $p < 0.001$). Error bars indicate SEM. Scale bar indicates 2 μm.

*2018*; *Latorre et al., 2017*). Thus, Shank mutations could broadly alter neuron and muscle function via changes in CaV-BK coupling. It will be very interesting to determine if this new function for Shank is conserved in other animals, including humans.

## SHN-1 promotes microdomain coupling of CaV1 and BK channels

BK channel activation requires tight coupling to CaV channels (*Berkefeld et al., 2006*). Our results suggest that SHN-1 promotes CaV1-BK microdomain coupling. APs were prolonged in *shn-1* (null), *shn-1*(ΔPDZ), and *egl-19*(ΔVTTL) mutants and in all cases these defects were eliminated in double mutants lacking SLO-2. Interestingly, although all impair SLO-2 mediated AP repolarization, these mutations had distinct effects on SLO-2 channels. SLO-2 currents were reduced in *shn-1* (null) and *shn-1*(ΔPDZ) mutants but were unaffected in *egl-19*(ΔVTTL) mutants. SLO-2 puncta intensity was decreased in *shn-1*(null) mutants but was unaffected in *shn-1*(ΔPDZ) and *egl-19*(ΔVTTL) mutants. These differences suggest that these mutants comprise an allelic series for EGL-19 to SLO-2 coupling defects in the following hierarchy *shn-1* (null) >*shn-1*(ΔPDZ) > *egl-19*(ΔVTTL) mutants. Based on these results, we propose that multiple protein interactions progressively tighten CaV-BK coupling. Specifically, we propose that: (1) multiple SHN-1 domains act together to promote SLO-2 coupling to EGL-19, accounting for the distinct phenotypes observed in *shn-1*(null) and *shn-1*(ΔPDZ) mutants; (2) SHN-1 promotes formation (or stability) of SLO-2 clusters in the plasma membrane, as indicated by decreased SLO-2 puncta intensity in *shn-1*(null) mutants; (3) beyond this trafficking function, SHN-1's PDZ domain tightens SLO-2 coupling to nearby calcium channels, accounting for the smaller SLO-2 current but unaltered SLO-2 puncta intensity in *shn-1*(ΔPDZ) mutants; (4) SHN-1 PDZ binding to EGL-19's c-terminus promotes rapid SLO-2 activation during APs, accounting for the increased AP width but unaltered SLO-2 current and puncta intensity in *egl-19*(ΔVTTL) mutants; and (5) SHN-1's PDZ must bind multiple proteins (not just EGL-19) to promote SLO-2 activation, accounting for the different phenotypes found in in *shn-1*(ΔPDZ) and *egl-19*(ΔVTTL) mutants. Multivalent interactions between scaffolds and their client proteins may represent a general mechanism for promoting microdomain signaling. Although SHN-1 binds EGL-19, SHN-1 may not directly link EGL-19 to SLO/BK channels. Instead, SHN-1 may link EGL-19 to other proteins required for SLO channel localization, for example components of the dystrophin complex (*Kim et al., 2009*; *Sancar et al., 2011*).

## Implications for understanding neurodevelopmental disorders

Several studies suggest that Shank3 deletions and duplications are both linked to ASD and schizophrenia, suggesting that opposite changes in Shank3 levels produce similar or overlapping psychiatric phenotypes (*Bonaglia et al., 2006*; *Durand et al., 2007*; *Failla et al., 2007*; *Gauthier et al., 2010*; *Han et al., 2013*). It remains possible that more detailed analysis will reveal phenotypic differences between gain and loss of human Shank3. In either case, it is currently unclear how opposite changes in Shank3 levels produce psychiatric phenotypes. Different (potentially opposite) biochemical defects arising from decreased and increased Shank dosage could produce psychiatric traits, perhaps by circuit level mechanisms (*Antoine et al., 2019*; *Peixoto et al., 2016*). For example, Shank duplications and hemizygosity could act in different cells or circuits to produce psychiatric traits. Our results provide support for a second possibility. We find that increased and decreased *shn-1* gene dosage produce similar cell autonomous CaV1-BK coupling defects. Two prior studies suggest that bidirectional changes in Shank produce similar defects in Wnt signaling and CaV1 current density (*Harris et al., 2016*; *Pym et al., 2017*). Collectively, these results suggest that some biochemical functions of Shank exhibit this unusual pattern of dose sensitivity and consequently could contribute to the pathophysiology of human Shankopathies (i.e. Shank3 mutations, CNVs, or PMS).

The role of human Shank in CaV1-BK coupling has not been tested. Nonetheless, it seems plausible that this new physiological function could contribute to neuropsychiatric or co-morbid phenotypes associated with human Shankopathies. Consistent with this idea, PMS and human KCNMA1/BK mutations are associated with several shared phenotypes including: autism, developmental delay, intellectual disability, hypotonia, seizures, and gastrointestinal defects (i.e. vomiting, constipation, or diarrhea) (*Bailey et al., 2019*; *Laumonnier et al., 2006*; *Phelan and McDermid, 2012*; *Soorya et al., 2013*; *Witmer et al., 2019*). Moreover, BK channels are regulated by two high confidence ASD genes (UBE3A and hnRNP U). BK channels are degraded by the ubiquitin ligase UBE3A (*Sun et al., 2019*), mutations in which cause Angelman's syndrome. The RNA binding protein hnRNP U promotes

translation of *slo-2* mRNA (*Liu et al., 2018*). Collectively, these results support the idea that disrupted CaV1-BK channel coupling could play an important role in shankopathies and that BK channels may represent an important therapeutic target for treating these disorders.

# Materials and methods

## Key resources table

| Reagent type (species) or resource | Designation | Source or reference | Identifiers | Additional information |
|---|---|---|---|---|
| Strain, strain background (*C. elegans*) | N2 Bristol | https://cgc.umn.edu/ | N2 | Wild-type reference |
| Strain, strain background (*C. elegans*) | *slo-1(js379)* | *Wang et al., 2001* | NM1968 | |
| Strain, strain background (*C. elegans*) | *slo-2(nf100)* | *Santi et al., 2003* | LY100 | |
| Strain, strain background (*C. elegans*) | *slo-1(js379);slo-2(nf100)* | This paper | KP10046 | |
| Strain, strain background (*C. elegans*) | *shk-1(ok1581)* | *Liu et al., 2011* | RB1392 | |
| Strain, strain background (*C. elegans*) | *shk-1(ok1581);slo-2(nf100)* | This paper | KP10879 | |
| Strain, strain background (*C. elegans*) | *shn-1(tm488)* | *Oh et al., 2011* | KP7032 | |
| Strain, strain background (*C. elegans*) | *shn-1(nu712)* | This paper | KP10151 | |
| Strain, strain background (*C. elegans*) | *shn-1(nu697)* | This paper | KP10082 | |
| Strain, strain background (*C. elegans*) | nuSi26 | *Pym et al., 2017* | KP7493 | Pmyo-3::shn-1A MOSSci |
| Strain, strain background (*C. elegans*) | nuSi572 | This paper | KP10696 | Muscle CRE |
| Strain, strain background (*C. elegans*) | nuSi502 | This paper | KP10497 | Pan neuron CRE |
| Strain, strain background (*C. elegans*) | *shn-1(nu697);nuSi572* | This paper | KP10767 | |
| Strain, strain background (*C. elegans*) | *shn-1(nu697);nuSi502* | This paper | KP10768 | |
| Strain, strain background (*C. elegans*) | nuSi205 | This paper | KP9234 | Ubiquitous GFP$_{1-10}$ |
| Strain, strain background (*C. elegans*) | *shn-1(nu604 GFP$_{11}$)* | This paper | KP8587 | |
| Strain, strain background (*C. elegans*) | *shn-1(nu652); nuSi205* | This paper | KP9548 | |
| Strain, strain background (*C. elegans*) | nuSi470 | This paper | KP10393 | Muscle CRE |
| Strain, strain background (*C. elegans*) | *shn-1(nu652);nuSi470* | This paper | KP10437 | |
| Strain, strain background (*C. elegans*) | *shn-1(nu604);nusi205* | This paper | KP9232 | |
| Strain, strain background (*C. elegans*) | *shn-1(nu542ΔPDZ)* | This paper | KP9898 | |
| Strain, strain background (*C. elegans*) | *egl-19(nu496ΔVTTL)* | *Pym et al., 2017* | KP7992 | |
| Strain, strain background (*C. elegans*) | *egl-19(nu496);shn-1(tm488)* | *Pym et al., 2017* | KP8046 | |
| Strain, strain background (*C. elegans*) | nuSi144 | This paper | KP9814 | muscle GFP$_{1-10}$ |
| Strain, strain background (*C. elegans*) | *slo-2(nu725 GFP$_{11}$)* | This paper | KP10285 | |
| Strain, strain background (*C. elegans*) | *slo-2(nu725);nuSi144* | This paper | KP10031 | |
| Strain, strain background (*C. elegans*) | *shn-1(nu712); slo-2(nu725);nuSi144* | This paper | KP10894 | |
| Strain, strain background (*C. elegans*) | *shn-1(nu542); slo-2(nu725);nuSi144* | This paper | KP10890 | |
| Strain, strain background (*C. elegans*) | *egl-19(nu496); slo-2(nu725);nuSi144* | This paper | KP10891 | |
| Strain, strain background (*C. elegans*) | *slo-1(nu678 GFP$_{11}$)* | This paper | KP9826 | |
| Strain, strain background (*C. elegans*) | *slo-1(nu678);nuSi144* | This paper | KP10030 | |
| Strain, strain background (*C. elegans*) | *shn-1(nu712); slo-1(nu678);nuSi144* | This paper | KP10892 | |
| Strain, strain background (*C. elegans*) | *shn-1(nu712);slo-2(nf100)* | This paper | KP10880 | |
| Strain, strain background (*C. elegans*) | *shn-1(nu542);slo-2(nf100)* | This paper | KP10881 | |
| Strain, strain background (*C. elegans*) | *egl-19(nu496);slo-2(nf100)* | This paper | KP10882 | |
| Strain, strain background (*C. elegans*) | nuSi458 | This paper | KP10374 | muscle Cherry$_{1-10}$ SL2 GFP$_{1-10}$ |

*Continued on next page*

*Continued*

| Reagent type (species) or resource | Designation | Source or reference | Identifiers | Additional information |
|---|---|---|---|---|
| Strain, strain background (*C. elegans*) | *egl-19(nu722* Cherry₁₁) | This paper | KP10230 | |
| Strain, strain background (*C. elegans*) | *slo-2(nu725);egl-19(nu722);nusi458* | This paper | KP10816 | |
| Strain, strain background (*C. elegans*) | *shn-1(nu712); slo-2(nu725);egl-19(nu722);nusi458* | This paper | KP10816 | |
| Strain, strain background (*C. elegans*) | *shn-1(nu712);vsls48* | This paper | KP10883 | vsls48 is Punc-17::GFP |
| Strain, strain background (*C. elegans*) | *slo-2(nf100);vsls48* | This paper | KP10884 | |
| Strain, strain background (*C. elegans*) | *shn-1(nu712);slo-2(nf100);vsls48* | This paper | KP10885 | |
| Strain, strain background (*C. elegans*) | *slo-2(nu725);nuSi144* | This paper | KP10886 | |
| Strain, strain background (*C. elegans*) | *shn-1(nu712);slo-2(nu725);nusi144* | This paper | KP10887 | |
| Strain, strain background (*C. elegans*) | *slo-1(nu678); slo-2(nf100); nuSi144* | This paper | KP10895 | |
| Strain, strain background (*C. elegans*) | *slo-2(nu725); slo-1(js379);nuSi144* | This paper | KP10896 | |
| Strain, strain background (*C. elegans*) | *slo-2(nu725);nuSi250* | This paper | KP10897 | nuSi250 is Punc-129 GFP₁₋₁₀ |
| Strain, strain background (*C. elegans*) | *shn-1(nu712); slo-2(nu725 GFP11);nuSi250* | This paper | KP10898 | |
| Strain, strain background (*C. elegans*) | *egl-19(nu496); shn-1(nu712)* | This paper | KP10906 | |
| Strain, strain background (*C. elegans*) | *unc-68(nu664); nuSi144* | This paper | KP9802 | |
| Strain, strain background (*C. elegans*) | *unc-68(nu664); shn-1(tm488); nuSi144* | This paper | KP10040 | |
| Strain, strain background (*C. elegans*) | *shn-1(nu604,nu652); nuSi502* | This paper | KP10907 | |
| Strain, strain background (*E. coli*) | OP50 | *Brenner, 1974* | OP50 | Worm food |
| Sequence-based reagent | egl-19 residue 2 | This paper | N/A | TTACCTGACATGATGGACAC |
| Sequence-based reagent | shn-1 ΔPDZ 5' | This paper | N/A | gtgattccacgtggtgtcaa |
| Sequence-based reagent | shn-1 ΔPDZ 3' | This paper | N/A | gtagctgatatgagtagggg |
| Sequence-based reagent | shn-1 intron one loxP insertion | This paper | N/A | tcaatttcagAAGTTCCTTG |
| Sequence-based reagent | shn-1 3' UTR loxP insertion | This paper | N/A | gaaaaggcatagaatcagtg |
| Sequence-based reagent | shn-1 intron two insertion for STOP cassette | This paper | N/A | ggggaaagatatgcatctga |
| Sequence-based reagent | shn-1 residue 946 insertion | This paper | N/A | CACATCTTCTCGAACGTCAC |
| Sequence-based reagent | slo-1 residue 1,121 insertion | This paper | N/A | cccggctcgtactccagtcc |
| Sequence-based reagent | slo-2 residue 1,092 insertion | This paper | N/A | ctgcgtcttagaccccttct |
| Recombinant DNA reagent | Pmyo-3::gfp₁₋₁₀ | This paper | KP#3,315 | muscle GFP₁₋₁₀ |
| Recombinant DNA reagent | Peft-3::gfp₁₋₁₀ | This paper | KP#4,524 | ubiquitous GFP₁₋₁₀ |
| Recombinant DNA reagent | Punc-129::gfp₁₋₁₀ | This paper | KP#4,525 | GFP₁₋₁₀ in DA/B neurons |
| Recombinant DNA reagent | Ppat-10 Cherry₁₋₁₀ SL2 GFP ₁₋₁₀ | This paper | KP#4,526 | Cherry₁₋₁₀ and GFP₁₋₁₀ in muscles |
| Recombinant DNA reagent | Pmyo-3::CRE | This paper | KP#4,527 | muscle CRE |
| Recombinant DNA reagent | *Psbt-1::CRE* | This paper | KP#4,528 | Pan neuron CRE |
| Recombinant DNA reagent | *Peft-3::CRE* | This paper | KP#4,529 | germline CRE |
| Chemical compound, drug | Nemadipine-A | Abcam | ab145991 | N/A |
| Software, algorithm | MATLAB R2018a | MATLAB | N/A | N/A |
| Software, algorithm | Fiji | https://fiji.sc/ | N/A | N/A |
| Software, algorithm | ClampFit | Molecular Devices | N/A | N/A |
| Software, algorithm | Prism 9 | GraphPad | N/A | N/A |
| Software, algorithm | Origin 2019 | OriginLab | N/A | N/A |
| Software, algorithm | Adobe illustrator 2020 | Adobe | N/A | N/A |

**Table 2.** Alleles used in this study.

| Allele: | Description: | Reference: |
|---|---|---|
| *shn-1(tm488)* | 1537 nt deletion, frameshift at codon 118 | *Oh et al., 2011* |
| *shn-1(nu697)* | LoxP sites in intron 1 and 3'UTR | This study |
| *shn-1(nu712)* | derived by germline CRE recombination of *nu697* | This study |
| *shn-1(nu652)* | stop cassette (flanked by FLEX sites) in intron two in "OFF" orientation | This study |
| *shn-1(nu600* GFP$_{11}$) | seven copies GFP$_{11}$ inserted at codon 945 of SHN-1A | This study |
| *shn-1(nu542ΔPDZ)* | deletes aa 446–532 of SHN-1A | This study |
| *egl-19(nu722* Cherry$_{11}$) | six copies sfCherry$_{11}$ inserted at codon 2 | This study |
| *egl-19(nu496ΔVTTL)* | WT C-term PAENSSRQHDSRGGSQEDLLLVTTL replaced with PMIHAEDHKKSYF | *Pym et al., 2017* |
| *unc-68(nu664* GFP$_{11}$) | seven copies GFP$_{11}$ inserted at codon 3,705 of UNC-68A | *Piggott et al., 2021* |
| *slo-2(nu725* GFP$_{11}$) | seven copies GFP$_{11}$ inserted at codon 1,092 of SLO-2A | This study |
| *slo-1(nu678* GFP$_{11}$) | seven copies GFP$_{11}$ inserted at codon 1,130 of SLO-1A | This study |
| *slo-1(js379)* | Q251stop | *Wang et al., 2001* |
| *slo-2(nf100)* | in frame deletion of aa 450–569 | *Santi et al., 2003* |
| *shk-1(ok1581)* | P253stop | *Liu et al., 2011* |

## Strains

Strain maintenance and genetic manipulation were performed as described (*Brenner, 1974*). Animals were cultivated at room temperature (~22 °C) on agar nematode growth media seeded with OP50 bacteria. Alleles used in this study are described in *Table 2* and are identified in each figure legend. All strains utilized are listed in the Key Resources Table. Transgenic animals were prepared by micro-injection, and integrated transgenes were isolated following UV irradiation, as described (*Dittman and Kaplan, 2006*). Single copy transgenes were isolated by the MoSCI and miniMoS techniques (*Frøkjaer-Jensen et al., 2008*; *Frøkjær-Jensen et al., 2014*).

## *Shn-1* dosage experiments

Animals with different *shn-1* copy numbers were constructed as follows: 0 copies, *shn-1(nu712)* homozygotes; one copy, *unc-17::gfp (LX929)* males were crossed with *shn-1(nu712)* homozygotes and *gfp*-expressing hermaphrodites were analyzed; two copies, WT were analyzed; four copies, WT animals homozygous for the single copy transgene expressing SHN-1A in body muscles (*nuSi26*). The *nuSi26* transgene was described in our earlier study (*Pym et al., 2017*).

## CRISPR alleles

CRISPR alleles were isolated as described (*Arribere et al., 2014*). Briefly, we used *unc-58* as a co-CRISPR selection to identify edited animals. Animals were injected with two guide RNAs (gRNAs) and two repair templates, one introducing an *unc-58* gain of function mutation and a second modifying a gene of interest. Progeny exhibiting the *unc-58(gf)* uncoordinated phenotype were screened for successful editing of the second locus by PCR. Split GFP and split sfCherry constructs are described in *Feng et al., 2017*. MiniMOS inserts in which Pmyo-3 drives expression of either GFP$_{1-10}$ (nuSi144) or sfCherry$_{1-10}$ SL2 GFP$_{1-10}$ (nuSi458) were created.

Tissue specific *shn-1* knockout was performed by introducing LoxP sites into intron 1 and the 3'UTR of the endogenous locus, in *shn-1(nu697)*, and expressing CRE in muscles (*pat-10* promoter) or neurons (*sbt-1* promoter). Tissue-specific *shn-1* rescue was performed by introducing a stop cassette into intron 2 of *shn-1* using CRISPR, creating the *shn-1(nu652)* allele. The stop cassette consists of a synthetic exon (containing a consensus splice acceptor sequence and stop codons in all reading frames) followed by a 3' UTR and transcriptional terminator taken from the *flp-28* gene (the 564 bp sequence just 3' to the *flp-28* stop codon). The stop cassette is flanked by FLEX sites (which are modified loxP sites that mediate CRE induced inversions) (*Schnütgen and Ghyselinck, 2007*). In this manner, orientation of the stop cassette within the *shn-1* locus is controlled by CRE expression. Expression of *shn-1* is reduced when the stop cassette is in the OFF configuration (i.e. the same

orientation as *shn-1*) but is unaffected in the ON configuration (opposite orientation). The endogenous *flp-28* gene is located in an intron of W07E11.1 (in the opposite orientation). Consequently, we reasoned that the *flp-28* transcriptional terminator would interfere with *shn-1* expression in an orientation selective manner. A similar strategy was previously described for conditional gene knockouts in *Drosophila* (**Fisher et al., 2017**).

## Fluorescence imaging

Worms were immobilized on 10% agarose pads with 0.3 µl of 0.1 µm diameter polystyrene microspheres (Polysciences 00876–15, 2.5% w/v suspension). Body muscles just anterior to the vulva were imaged. Images were taken with a Nikon A1R confocal, using a 60 X/1.49 NA oil objective, with Nyquist sampling. Image volumes spanning the muscle surface were collected (~10 planes/volume, 0.15 µm between planes, and 0.06 µm /pixel). Maximum intensity projections for each volume were auto-thresholded, and puncta were identified as round fluorescent objects (area >0.1 µm$^2$), using analysis of particles. Mean fluorescent intensity in each punctum was analyzed in the raw images. All image analysis was done using FIJI.

## Electrophysiology

Whole-cell patch-clamp measurements were performed using a Axopatch 200B amplifier with pClamp 10 software (Molecular Devices). The data were sampled at 10 kHz and filtered at 5 kHz. All recordings were performed at room temperature (~19°C–21°C).

Muscle AP recordings- The bath solution contained (in mM): NaCl 140, KCl 5, $CaCl_2$ 5, $MgCl_2$ 5, dextrose 11 and HEPES 5 (pH 7.2, 320 mOsm). The pipette solution contained (in mM): Kgluconate 120, KOH 20, Tris 5, $CaCl_2$ 0.25, $MgCl_2$ 4, sucrose 36, EGTA 5 (or BAPTA 5), and $Na_2ATP$ 4 (pH 7.2, 323 mOsm). Spontaneous APs were recorded in current-clamp without current injection. Cell resistance ($R_{in}$) was measured following a 10 pA pulse injection. AP traces were analyzed in Matlab. APs were defined as depolarizations lasting <150ms. PPs were defined as depolarizations lasting >150ms.

$K^+$ current recordings - The bath solution contained (in mM): NaCl 140, KCl 5, $CaCl_2$ 5, $MgCl_2$ 5, dextrose 11 and HEPES 5 (pH 7.2, 320 mOsm). For $Ik_{loCl}$ recordings, the pipette solution contained (in mM): Kgluconate 120, KOH 20, Tris 5, $CaCl_2$ 0.25, $MgCl_2$ 4, sucrose 36, EGTA 5, and $Na_2ATP$ 4 (pH 7.2, 323 mOsm). For $Ik_{hiCl}$ recordings, the pipette solution contained (in mM): KCl 120, KOH 20, Tris 5, $CaCl_2$ 0.25, $MgCl_2$ 4, sucrose 36, EGTA 5 (or BAPTA 5), and $Na_2ATP$ 4 (pH 7.2, 323 mOsm). The voltage-clamp protocol consisted of –60 mV for 50ms, –90 mV for 50ms, test voltage (from –60 mV to +60 mV) 150ms. The repetitive stimulus protocol was –20 mV for 50ms, + 30 mV for 50ms, which was repeated 20 times. In figures, we show outward currents evoked at +30 mV, which corresponds to the peak amplitude of muscle APs. In some recordings, EGL-19 channels were blocked by adding 5 µM nemadipine to the pipette solution. Patch clamp recording of $Ik_{hiCl}$ in ACh motor neurons was done using solutions described above for the muscle recordings. ACh neurons were identified for patching by expression of an *unc-17* transcriptional reporter (P*unc-17*::GFP).

## Statistical methods

For normally distributed data, significant differences were assessed with unpaired t tests (for two groups) or one way ANOVA with post-hoc Dunn's multiple comparisons test (for >2 groups). For non-normal data, differences were assessed by Mann-Whitney (two groups) or Kruskal-Wallis test with post-hoc Dunn's multiple comparisons test ( > 2 groups). Data graphing and statistics were performed in GraphPad Prism 9. No statistical method was used to select sample sizes. Data shown in each figure represent contemporaneous measurements from mutant and control animals over a period of 1–2 weeks. For electrophysiology, data points represent mean values for individual neuron or muscle recordings (which were considered biological replicates). For imaging studies, data points represent mean puncta fluorescence values in individual animals (which were considered biological replicates). All data obtained in each experiment were analyzed, without any exclusions.

## Acknowledgements

We thank the following for strains, advice, reagents, and comments on the manuscript: *C. elegans* genetics stock center (CGC), S Mitani, and members of the Kaplan lab. This work was supported by an

NIH research grant to JK (NS32196). The CGC is funded by the NIH Office of Research Infrastructure Programs (P40 OD010440).

## Additional information

### Funding

| Funder | Grant reference number | Author |
| --- | --- | --- |
| National Institutes of Health | NS32196 | Joshua M Kaplan |

The funders had no role in study design, data collection and interpretation, or the decision to submit the work for publication.

### Author contributions

Luna Gao, Conceptualization, Data curation, Formal analysis, Funding acquisition, Investigation, Writing - original draft, Writing – review and editing; Jian Zhao, Conceptualization, Data curation, Formal analysis, Investigation, Writing – review and editing; Evan Ardiel, Qi Hall, Formal analysis, Investigation, Methodology, Software, Writing – review and editing; Stephen Nurrish, Conceptualization, Investigation, Writing – review and editing; Joshua M Kaplan, Conceptualization, Funding acquisition, Investigation, Writing - original draft, Writing – review and editing

### Author ORCIDs
Stephen Nurrish (iD) http://orcid.org/0000-0002-2653-9384
Joshua M Kaplan (iD) http://orcid.org/0000-0001-7418-7179

### Decision letter and Author response
Decision letter https://doi.org/10.7554/eLife.75140.sa1
Author response https://doi.org/10.7554/eLife.75140.sa2

## Additional files

### Supplementary files
• Transparent reporting form

### Data availability
All data generated or analyzed in this study are included in the manuscript.

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
