## [Editor Report]

Mutations altering the scaffolding protein Shank are linked to several psychiatric disorders. Here the authors take advantage of *C. elegans* genetics and muscle physiology to demonstrate that Shank binds CaV1 voltage activated calcium channels and promotes CaV1 coupling to calcium activated potassium channels.

---

## [Decision Letter]

**Decision letter after peer review:**

Thank you for submitting your article "Shank promotes action potential repolarization by recruiting BK channels to calcium nanodomains" for consideration by *eLife*. Your article has been reviewed by 3 peer reviewers, including Graeme W Davis as Reviewing Editor and Reviewer #1, and the evaluation has been overseen by Piali Sengupta as the Senior Editor.

Essential revisions:

Summary: The authors have assembled a compelling study annotating the function of *C. elegans* Shank protein at discrete specializations in skeletal muscle, acting to couple calcium influx to BK channel function. Shank mutations, by altering the organization of these specializations, lead to altered calcium-driven action potential waveforms and muscle activation. The reviewers consider the work well executed and nicely presented. Major concerns address a few outstanding issues to clarify or strengthen specific points in the manuscript, all of which should be easily achieved. Additional critiques are focused on text revision to enhance clarity and acknowledge specific issues. Again, these are straightforward suggestions. The authors should be congratulated on a nice study.

Major Concerns:

1. Figures 2C and D: using floxed (nu697) or flexed (nu652) shn-1 alleles, they show that expressing CRE in muscles enhances PP rate and AP width, while expressing CRE in neurons has no effect on PP rate and a moderate, but not significant effect on AP width. Could it be that shn-1 affects muscle AP through a function in both tissues? This may be addressed by expressing CRE simultaneously in neurons and muscle to test if PP rate or AP width are further enhanced/rescued. Also, it appears that the authors omitted the control expressing CRE in neurons of nu652, which will strengthen the conclusions.

2. Figure 3 could be strengthened by including analysis of ∆VTTL; ∆PDZ double mutants. The findings will support that these two domains indeed interact with each other, as opposed to the idea that each domain mediates a different task, each incrementally affecting AP repolarization.

3. Figure 5A-B implies potassium currents are mediated by 2 sources, one channel composed of SLO-1/SLO-2 and another homomeric SLO-2. The finding in Figure 5C-D that SLO-2 signals are reduced in the absence of SLO-1 implies that the remaining GFP signals may represent SLO-2 homomers. If SLO-1 is exclusively heterodimer with SLO-2, how do the authors explain the observation that SLO-1 signals are unchanged in slo-2 mutants? Based on the data, it remains possible that functioning SLO-1 containing channels are not required for repolarization and instead, SLO-1 is only required for SLO-2 trafficking or localization. The statement that "..these results suggest that rapid muscle repolarization following Aps is mediated by SLO1/2 heteromeric channels." The authors should acknowledge that SLO-2 homomers may also support rapid repolarization.

4. Figure 6D-E: The authors suggest that SHN-1 stabilizes SLO to nearby EGL-19 channels to link Ca to K currents. However, they also observe that ∆VTTL does not affect SLO-2 currents. While the authors address this contradiction in their discussion, a test of the AP phenotypes in egl-19∆VTTL; slo-2 double mutants compared to the singles (related to point 2 about the Figure 3 analysis) would help rule out any SLO-2 independent role for the egl-19 VTTL domain.

5. Table 2 is a list of alleles used or generated. The authors should also include a comprehensive list of strains used in this study.

6. Figures 2 and 3. The only data in the manuscript that are less than completely compelling are the assessments of PP rate where a subset of recordings seem to be affected and the changes are generally less than 1/10th of a Hz. The effect seems to reach high levels of significance due to the fact that the rate is near zero in controls (as well as a large number of the mutant recordings). There is a disruption, but the relevance here to muscle and animal function is a bit hard to understand. This should be clarified in the text and, perhaps, the authors could consider other aspects of the data that could be quantified – charge comes to mind.

7. Action potentials. These are slow, calcium-driven, muscle action potentials. The most appropriate reference point is probably cardiac muscle. The implicit assertion that there is direct relevance to sodium-driven action potentials in the CNS should be revised and broadened to non-neuronal muscle. This need not diminish the relevance of the work – in fact it may broaden the relevance.

8. The emphasis on nano-domain organization is problematic. As I understand, the sites of protein co-localization represent large micro-domains within the muscle t-tubule system. Obviously, nano-domains are relevant in mammalian neurons to couple calcium entry to BK channel function in both time and space during the sub-millisecond kinetics of the sodium action potential. The tight temporal coupling demands physical coupling. Work at the frog NMJ, imaging calcium domains, argues that the size of calcium domains at sites of entry are approx. 1.5µm and this size is not influenced significantly by the presence of EGTA. Given the much slower temporal dynamics of the *C. elegans* muscle action potential, and the large size of the protein micro-domains, I do not fully understand the argument for nano-domain coupling. This should probably be revised.

9. The muscle sites seem to be coincident with T-tubule domains. Can the authors provide evidence of other resident markers that do, or do not change? This will provide more general relevance and give a sense of whether there is broad-based disorganization of these sites versus specific effects on the channels focused on in the text.

10. The authors imply that shank, CaV1, and the Slo's form a functional complex in the muscle membrane, but the data are somewhat indirect. It would be useful if the authors could spell this out better so the reader can assess the strength of this inference. Is there direct evidence (pull-downs, etc) for such a 3-way complex?*Reviewer #1 (Recommendations for the authors):*

Technically, the paper from the Kaplan rests on solid ground. An array of mutations and transgenic lines are used in the study the Shank gene, and are nicely documented. The electrophysiological assessment of ionic currents in *C. elegans* muscle is clear and well documented. Finally, the light-level protein localization analyses are clear and well documented. The authors cleanly define a set of phenotypes caused by mutations in the Shank gene, influencing muscle action potentials in *C. elegans*.

*Reviewer #2 (Recommendations for the authors):*

The authors use *C. elegans* to explore the relationship between shank, CaV1 (Ca) channels, and BK (slo) calcium-dependent K channels in controlling muscle excitability. They use a range of genetic approaches to mutate or knock out one or more of these players and assess the impact on muscle action potential generation and slo currents. Their data show that shank controls AP width and plateau potential generation (pp) through Slo channels, and this effect is cell-autonomous in muscle. They go on to suggest this effect is mediated through CaV1-slo coupling, by using previously characterized mutant that reduce this binding. Because these mutations may affect binding with other partners, these experiments do not unequivocally implicate direct binding between these three players; however, use of fast and slow Ca buffers also suggests that shank keeps CaV1 and slo in close association, presumably allowing Ca influx through CaV1 channels to effectively activate slo channels. Finally, they show that overexpression of shank has a similar impact on excitability as reduction; this is somewhat puzzling and not further explored mechanistically, but is interesting given gene dosage effects of shanks in humans.

*Reviewer #3 (Recommendations for the authors):*

Gao et al., present a nice set of data, using electrophysiology and molecular genetics, to address the function of *C. elegans* Shank (shn-1) in shaping muscle action potentials. Using genome-edited Cre-dependent deletion and expression of SHN-1, they show that removal of shn-1 specifically in body muscle widens the duration of action potentials and increases prolonged depolarization events known as plateau potentials (PP). They provide new evidence that SHN-1 couples the activity of the calcium channel EGL-19 to that of the BK potassium channels SLO-1/2. They additionally reveal that action potentials are sensitive to SHN-1 dosage. The experiments are generally conducted rigorously, and conclusions are stated appropriately. The findings offer insights into how human Shank misexpression might contribute to neurological disorder.

---

## [Author Response]

Essential revisions:Major Concerns:1. Figures 2C and D: using floxed (nu697) or flexed (nu652) shn-1 alleles, they show that expressing CRE in muscles enhances PP rate and AP width, while expressing CRE in neurons has no effect on PP rate and a moderate, but not significant effect on AP width. Could it be that shn-1 affects muscle AP through a function in both tissues? This may be addressed by expressing CRE simultaneously in neurons and muscle to test if PP rate or AP width are further enhanced/rescued. Also, it appears that the authors omitted the control expressing CRE in neurons of nu652, which will strengthen the conclusions.

As requested, we now analyze AP firing patterns following neuronal CRE expression in *nu652* (Figure 2). These data are described in the revised text, as follows:

Page 8:

“By contrast, *shn-1*(neuron KO) and *shn-1*(neuron rescue) had no effect on PP rate or AP widths (Figure 2C-D).”

2. Figure 3 could be strengthened by including analysis of ∆VTTL; ∆PDZ double mutants. The findings will support that these two domains indeed interact with each other, as opposed to the idea that each domain mediates a different task, each incrementally affecting AP repolarization.

To address this concern, we now analyze ΔVTTL; *shn-1(null)* double mutants. Because the *shn-1* null has a stronger SLO channel defect than the ΔPDZ mutant, we thought it would be better to analyze this double mutant. These new results are described as follows:

Page 8 (results):

“Furthermore, the *shn-1(nu712* null*)* and *egl-19(nu496* ΔVTTL*)* mutations did not have additive effects on PP rate and AP widths in double mutants (Figure 3, supplement 2).”

3. Figure 5A-B implies potassium currents are mediated by 2 sources, one channel composed of SLO-1/SLO-2 and another homomeric SLO-2. The finding in Figure 5C-D that SLO-2 signals are reduced in the absence of SLO-1 implies that the remaining GFP signals may represent SLO-2 homomers. If SLO-1 is exclusively heterodimer with SLO-2, how do the authors explain the observation that SLO-1 signals are unchanged in slo-2 mutants? Based on the data, it remains possible that functioning SLO-1 containing channels are not required for repolarization and instead, SLO-1 is only required for SLO-2 trafficking or localization. The statement that "..these results suggest that rapid muscle repolarization following Aps is mediated by SLO1/2 heteromeric channels." The authors should acknowledge that SLO-2 homomers may also support rapid repolarization.

The text was revised as recommended:

Page 11:

“Collectively, these results suggest that rapid muscle repolarization following APs is mediated by SLO-1/2 heteromeric channels and by SLO-2 homomers.”

4. Figure 6D-E: The authors suggest that SHN-1 stabilizes SLO to nearby EGL-19 channels to link Ca to K currents. However, they also observe that ∆VTTL does not affect SLO-2 currents. While the authors address this contradiction in their discussion, a test of the AP phenotypes in egl-19∆VTTL; slo-2 double mutants compared to the singles (related to point 2 about the Figure 3 analysis) would help rule out any SLO-2 independent role for the egl-19 VTTL domain.

The requested experiment was included in the original submission. These results are described in the text as follows:

Page 11:

“Consistent with this idea, AP widths in *slo-2* single mutants were not significantly different from those in *slo-2* double mutants containing *shn-1(nu712* null), *shn-1(nu542* ΔPDZ), or *egl-19(nu496* ΔVTTL) mutations (Figure 6A).”

5. Table 2 is a list of alleles used or generated. The authors should also include a comprehensive list of strains used in this study.

A complete list of strains utilized is now provided in the Key Resource Table.

6. Figures 2 and 3. The only data in the manuscript that are less than completely compelling are the assessments of PP rate where a subset of recordings seem to be affected and the changes are generally less than 1/10th of a Hz. The effect seems to reach high levels of significance due to the fact that the rate is near zero in controls (as well as a large number of the mutant recordings). There is a disruption, but the relevance here to muscle and animal function is a bit hard to understand. This should be clarified in the text and, perhaps, the authors could consider other aspects of the data that could be quantified – charge comes to mind.

We agree that identifying a physiological significance for the muscle plateau potentials (PPs) would be interesting. Experimentally, this would require a mutation (or drug) that blocks SHK-1/KCNA inactivation (thereby reducing PP rate). Given that we lack such a mutant (or drug), we have no data examining this issue; consequently, we would rather not speculate about this. Nonetheless, we believe that readers will find it helpful if the text comments on PPs and SHN-1’s impact on them because PPs are a salient feature of our AP recordings.

7. Action potentials. These are slow, calcium-driven, muscle action potentials. The most appropriate reference point is probably cardiac muscle. The implicit assertion that there is direct relevance to sodium-driven action potentials in the CNS should be revised and broadened to non-neuronal muscle. This need not diminish the relevance of the work – in fact it may broaden the relevance.

We agree that CaV-BK coupling is unlikely to alter sodium-driven APs; however, we cannot find anywhere in the text where this possibility is suggested (or implied). If the reviewer can point out a specific comment that should be revised, we will be happy to do so. Instead, we list several potential effects of altered CaV-BK coupling on neuron and muscle physiology:

Pages 16-17:

“Shank regulation of BK channels could have broad effects on neuron and muscle function. In neurons, BK channels are functionally coupled to CaV channels in the soma and dendrites, thereby regulating AP firing patterns and somatodendritic calcium transients Golding et al., 1999Storm, 1987(; ). In pre-synaptic terminals, BK channels limit the duration of calcium influx during APs, thereby decreasing neurotransmitter release Griguoli et al., 2016Yazejian et al., 2000(; ). In muscles, BK channels regulate AP firing patterns, calcium influx during APs, and muscle contraction Dopico et al., 2018Latorre et al., 2017(; ). Thus, Shank mutations could broadly alter neuron and muscle function via changes in CaV-BK coupling.”

8. The emphasis on nano-domain organization is problematic. As I understand, the sites of protein co-localization represent large micro-domains within the muscle t-tubule system. Obviously, nano-domains are relevant in mammalian neurons to couple calcium entry to BK channel function in both time and space during the sub-millisecond kinetics of the sodium action potential. The tight temporal coupling demands physical coupling. Work at the frog NMJ, imaging calcium domains, argues that the size of calcium domains at sites of entry are approx. 1.5µm and this size is not influenced significantly by the presence of EGTA. Given the much slower temporal dynamics of the *C. elegans* muscle action potential, and the large size of the protein micro-domains, I do not fully understand the argument for nano-domain coupling. This should probably be revised.

We thank the reviewers for this comment. As recommended, we replaced nano-domain with micro-domain throughout the text (including the title).

9. The muscle sites seem to be coincident with T-tubule domains. Can the authors provide evidence of other resident markers that do, or do not change? This will provide more general relevance and give a sense of whether there is broad-based disorganization of these sites versus specific effects on the channels focused on in the text.

As suggested, we now analyze SHN-1’s impact on three additional junctional markers (SLO-1, EGL-19, and UNC-68). These results are described as follows:

Page 13:

“Next, we asked if inactivating SHN-1 alters the localization of other muscle ion channels. SLO-1 puncta intensity was unaltered in *shn-1* null mutants, indicating that BK channels lacking SLO-2 were trafficked normally (Figure 7 supplement 2A-B). In body muscles, EGL-19/CaV1 channels are extensively co-localized with calcium channels in the endoplasmic reticulum (ER), UNC-68/Ryanodine Receptors (RYR) Piggott et al., 2021(). However, the puncta intensity of endogenous EGL-19(Cherry_11_) and UNC-68(GFP_11_)/RYR in body muscles were unaltered in *shn-1(nu712* null) mutants (Figure 7H and Figure 7 supplement 2C-D), suggesting that SHN-1 does not broadly regulate co-localization of ion channels at ER-plasma membrane junctional contacts.”

10. The authors imply that shank, CaV1, and the Slo's form a functional complex in the muscle membrane, but the data are somewhat indirect. It would be useful if the authors could spell this out better so the reader can assess the strength of this inference. Is there direct evidence (pull-downs, etc) for such a 3-way complex?

The text was revised as suggested:

Page 18:

“Although SHN-1 binds EGL-19, SHN-1 may not directly link EGL-19 to SLO/BK channels. Instead, SHN-1 may link EGL-19 to other proteins required for SLO channel localization, e.g. components of the dystrophin complex Kim et al., 2009Sancar et al., 2011(; ).”